# Regularized Behavior Cloning for Blocking the Leakage of Past Action Information

**Seokin Seo**[1], **HyeongJoo Hwang**[1], **Hongseok Yang**[1,2], **Kee-Eung Kim**[1,2]

[1]Kim Jaechul Graduate School of AI, KAIST
[2]School of Computing, KAIST
siseo@ai.kaist.ac.kr, hjhwang@ai.kaist.ac.kr,
hongseok.yang@kaist.ac.kr, kekim@kaist.ac.kr

## Abstract

For partially observable environments, imitation learning with observation histories (ILOH) assumes that control-relevant information is sufficiently captured in the observation histories for imitating the expert actions. In the offline setting where the agent is required to learn to imitate without interaction with the environment, behavior cloning (BC) has been shown to be a simple yet effective method for imitation learning. However, when the information about the actions executed in the past timesteps leaks into the observation histories, ILOH via BC often ends up imitating its own past actions. In this paper, we address this catastrophic failure by proposing a principled regularization for BC, which we name Past Action Leakage Regularization (PALR). The main idea behind our approach is to leverage the classical notion of conditional independence to mitigate the leakage. We compare different instances of our framework with natural choices of conditional independence metric and its estimator. The result of our comparison advocates the use of a particular kernel-based estimator for the conditional independence metric. We conduct an extensive set of experiments on benchmark datasets in order to assess the effectiveness of our regularization method. The experimental results show that our method significantly outperforms prior related approaches, highlighting its potential to successfully imitate expert actions when the past action information leaks into the observation histories.

## 1 Introduction

Imitation learning (IL) aims at learning a policy that recovers an expert's behavior from a demonstration dataset. Leveraging the information about state and expert action available in the dataset, IL has been successful in many real-world applications [4, 12, 26, 27, 32, 38, 43]. Although IL problems can be addressed using either online [17, 36] or offline algorithms [20, 33, 48], real-world tasks often impose restrictions on interacting with the environment due to safety, cost, and ethical concerns. Consequently, the practical necessity lies in the development of effective offline IL algorithms. Behavior cloning (BC) [33] is one of the most prominent offline IL algorithms, which learns to predict an expert's action for each given state using an offline expert dataset via supervised learning. BC has gained widespread recognition [7, 42] for providing a straightforward and effective solution, especially when full access to state information is available and the dataset is sufficiently extensive. However, when the state information is only partially available in an observation at each timestep, which is closer to the realistic scenario (e.g. an autonomous vehicle with a limited number

of sensors), training with any IL algorithm can be complicated. To enhance the ability of agent to infer missing control-relevant information from observations, incorporating the history of observations from adjacent past timesteps as input can be beneficial [2, 24].

When utilizing observation history for behavior cloning, a notable challenge emerges: the unnecessary dependence of the imitator's actions on the preceding actions, resulting in a suboptimal behavior in test time. In extreme cases, this dependency can lead to undesirable behaviors, such as the imitation policy that merely replicates its own actions from previous time steps [1], which leads to a catastrophic result particularly in safety-critical tasks like autonomous driving. This is often rooted at the inability to differentiate between actual and spurious causal relationship between observation features and expert actions within the collected data [8]. Past action information is a representative instance of such nuisance features, since it is a strongly correlated to the target expert action. This correlation can mislead the IL algorithm into recognizing past action information, which is not causally related to the target expert action, as a crucial feature for prediction. In this work, we refer to this misleading phenomenon as past action information leakage.

A natural way to mitigate this phenomenon is to adopt a mechanism that blocks the leakage of past action information irrelevant to the target expert action. In fact, Wen et al. [44] proposed an adversarial training approach to discard such redundant information in the representation of the observation history. This is achieved by maximizing the conditional entropy of the expert's previous action given the joint of the representation and target expert action. However, due to the intractability of direct computation of the conditional entropy, this approach relies on unstable adversarial learning. More recently, Chuang et al. [6] focuses on image-based observations and attempts to mitigate the past action dependency by splitting the policy representation into two parts: a representation of the observation history and a representation of the current observation. This method assumes a particular policy structure and does not provide a systematic general approach to block the leakage of past action information into the representation.

In this paper, we present a regularized behavior cloning framework that effectively mitigates the leakage of past action information. We formally define the problem of the leakage of past action information and establish a metric to quantify the magnitude of the leakage, employing the kernel-based method called HSCIC (Hilbert-Schmidt Conditional Independence Criterion) [30]. Building upon the metric, we devise an objective function that addresses the aforementioned problem, allowing for a more comprehensive understanding and interpretation of existing work [44]. Moreover, we propose a stable and efficient kernel-based regularization method that circumvents challenges such as adversarial learning, nested optimization and reliance on a neural estimator. Lastly, we conduct an extensive set of experiments which empirically show that our regularization method effectively blocks the leakage of past action information across a variety of control task benchmarks.

Our contributions are summarized as follows:

- We formally define the problem of the leakage of past action information based on the concept of conditional independence by quantifying the amount of leaked past action information.
- We introduce a principled framework for a behavior cloning with Past Action Leakage Regularization (PALR), which prevents the imitator from overfitting to leaked past information.
- We provide experimental results on established benchmarks, demonstrating the effectiveness of our method.

## 2   Related Work

**Invariant representation learning**   Learning representation invariant to any unwanted factors has been widely studied in various domains such as fair classification [22, 25, 37, 47], domain adaptation [11, 13, 18, 49], and imitation learning [6, 44]. One of the dominant approaches in invariant representation learning is adversarial learning [11, 13, 44, 47]. Adversarial learning algorithms

---

[1]The problem has many names: copycat problem [6, 44], inertia problem [7], latching effect [40]. Beyond the falsely leaded action repeating behavior, we introduce the past action information leakage to separate the reason and the result.

commonly train additional networks that predict the unwanted factors from the representations while enforcing representations to make those prediction models fail. Consequently, they require alternating optimization between the main and the additional models [13, 44, 47] or show numerical instability [11]. To bypass those shortcomings of adversarial learning, several information-theoretic approaches [18, 25] have been proposed. Based on Variational Auto-Encoders (VAE) [21], these methods proposed end-to-end learning algorithms that jointly optimize all of their components with numerical stability. However, these methods assume that the distribution of the representation is Gaussian, which restricts the flexibility of the representation. Wen et al. [46] concentrated on eliminating shortcuts in supervised learning by incorporating supplementary key information, demonstrating its enhancement of behavior cloning with observation histories. Recent methods have proposed to learn counterfactually or conditionally invariant representation, leveraging on kernel-based conditional independence metric [31, 35]. These methods demonstrated promising empirical results in synthetic domains, effectively mitigating the impact of nuisance correlations.

**Information leakage in imitation learning**   There is a growing understanding of the importance of addressing the correlation between expert actions and nuisance features, which are not essential for control and may even hinder performance. [8, 29, 39] Especially, termed as causal confusion [8], imitation learning exhibits a paradoxical phenomenon: having more information can lead to worse performance. Aligning with such observation, recent works have also demonstrated that accessing more information from the observation history leaks past action information so that the imitator may learn an undesirable policy that simply repeats the same action in the past [6, 7, 40, 44, 45]. To avoid learning degenerate solutions from the past action information, Wen et al. [44] proposed the regularization method based on the conditional entropy of the previous action given the representation of the history and the current action. However, their method involves a nested minimax optimization along with an additional neural network, which complicates the training process. We closely investigate their formulation in Section 4.2.1 to show their limitations as well as their connection to our method. Wen et al. [44] also introduced the action predictability metric to quantify the dependence between past action histories and imitator actions, relative to expert actions. However, this metric is not based on conditional independence, which is central to our argument concerning the past action leakage. Wen et al. [45] addressed this problem using a weighted behavior cloning method that upweights "keyframe" samples, which are more likely to be predicted from the action histories. Swamy et al. [40] showed that online interaction is both necessary and sufficient to resolve repeating behavior. However, in many real-world applications [4, 12, 26, 27, 32, 38, 43], online interaction is often infeasible due to safety, cost, and ethical considerations. In this work, we develop an offline algorithm that can robustly handle the leakage of past action information.

## 3   Preliminaries

### 3.1   Conditional independence

For random variables $X, Y$ which taking a value in $\mathcal{X}, \mathcal{Y}$ respectively, we write $P_X$, $P_{XY}$, and $P_{X|Y}$ for the marginal distribution of $X$, the joint distribution of $X$ and $Y$, and the conditional distribution of $X$ given $Y$, respectively. We say that random variables $X$ and $Y$ are conditionally independent given a random variable $Z$ or simply $Z$-conditionally independent if $P_{XY|Z}(x, y) = P_{X|Z}(x)P_{Y|Z}(y)$ for all $x, y \in \mathcal{X} \times \mathcal{Y}$. Also, we say that $X$ and $Y$ are independent if $P_{XY}(x, y) = P_X(x)P_Y(y)$ for all $x, y$. We denote $Z$-conditional independence by $X \perp\!\!\!\perp Y \mid Z$, and independence by $X \perp\!\!\!\perp Y$.

**Conditional Mutual Information (CMI)**   Mutual information (MI) is an information-theoretic quantity to measure the dependency between two random variables. MI between random variables $X$ and $Y$ is defined as $I(X;Y) = D_{\mathrm{KL}}(P_{XY}, P_X P_Y) = \mathbb{E}_{x,y \sim P_{XY}}[\log P_{XY}(x, y) - \log P_X(x)P_Y(y)]$. MI is always non-negative (i.e., $I(X;Y) \geq 0$), and it becomes zero if and only if $X \perp\!\!\!\perp Y$. Conditional mutual information (CMI) is defined similarly, but with conditional distributions. CMI between random variables $X$ and $Y$ given a random variable $Z$ is defined as $I(X;Y \mid Z) = \mathbb{E}_{P_Z}[D_{\mathrm{KL}}(P_{XY|Z}, P_{X|Z}P_{Y|Z})]$. As in the case of MI, $I(X;Y \mid Z) \geq 0$, and also $I(X;Y \mid Z) = 0$ if and only if $X \perp\!\!\!\perp Y \mid Z$.

**Hilbert-Schmidt Conditional Independence Criterion (HSCIC)**    Hilbert-Schmidt Independence Criterion (HSIC) [14] is a kernel-based measure that quantifies the dependency between two random variables. Let $\mathcal{H}_\mathcal{X}$ and $\mathcal{H}_\mathcal{Y}$ be reproducing kernel Hilbert spaces (RKHS) on $\mathcal{X}$ and $\mathcal{Y}$ with the corresponding reproducing kernels $k_\mathcal{X}$ and $k_\mathcal{Y}$, respectively. The tensor product $k_\mathcal{X} \otimes k_\mathcal{Y}$ of the kernels is a binary function on $\mathcal{X} \times \mathcal{Y}$ defined by $(k_\mathcal{X} \otimes k_\mathcal{Y})((x_1, y_1), (x_2, y_2)) := k_\mathcal{X}(x_1, x_2)k_\mathcal{Y}(y_1, y_2)$. The RKHS on $\mathcal{X} \times \mathcal{Y}$ with the kernel $k_\mathcal{X} \otimes k_\mathcal{Y}$ is called the tensor product RKHS of $\mathcal{H}_\mathcal{X}$ and $\mathcal{H}_\mathcal{Y}$, and it is denoted by $\mathcal{H}_\mathcal{X} \otimes \mathcal{H}_\mathcal{Y}$. HSIC between $X$ and $Y$ is defined to be the maximum mean discrepancy (MMD) between $P_{XY}$ and $P_X P_Y$ under $\mathcal{H}_\mathcal{X} \otimes \mathcal{H}_\mathcal{Y}$, i.e, the distance in the Hilbert space $\mathcal{H}_\mathcal{X} \otimes \mathcal{H}_\mathcal{Y}$ between the so-called kernel mean embeddings of $P_{XY}$ and $P_X P_Y$ to that space:

$$\text{HSIC}(X, Y) := \text{MMD}^2(P_{XY}, P_X P_Y; \mathcal{H}_\mathcal{X} \otimes \mathcal{H}_\mathcal{Y})$$
$$= \|\mu_{P_{XY}} - \mu_{P_X} \otimes \mu_{P_Y}\|^2_{\mathcal{H}_\mathcal{X} \otimes \mathcal{H}_\mathcal{Y}}$$

where $\mu_{P_{XY}}$ is the kernel mean embedding of the distribution $P_{XY}$ to $\mathcal{H}_\mathcal{X} \otimes \mathcal{H}_\mathcal{Y}$, i.e. $\mu_{P_{XY}}(x, y) := \mathbb{E}_{P_{XY}}[(k_\mathcal{X} \otimes k_\mathcal{Y})((X, Y), (x, y))]$, and $\mu_{P_X}$, $\mu_{P_Y}$ are defined similarly but using $\mathcal{H}_\mathcal{X}$, $\mathcal{H}_\mathcal{Y}$ instead. The $(\mu_{P_X} \otimes \mu_{P_Y})$ is simply the function in $\mathcal{H}_\mathcal{X} \otimes \mathcal{H}_\mathcal{Y}$ that maps $(x, y)$ to $\mu_{P_X}(x)\mu_{P_Y}(y)$. When $k_\mathcal{X} \otimes k_\mathcal{Y}$ satisfies some condition (i.e., characteristic), $\text{HSIC}(X, Y) = 0$ if and only if $X \perp\!\!\!\perp Y$ [14].

HSCIC (Hilbert-Schmidt Conditional Independence Criterion) is an extension of HSIC that quantifies the amount of conditional dependency between two random variables given another random variable. In this paper, we follow the definition of HSCIC based on conditional mean embedding [30], and use an estimator of HSCIC that draws samples from the joint distribution $P_{XYZ}$ and performs vector-valued RKHS regression. The conditional mean embedding of $X$ given $Z$ is a function in the RKHS $\mathcal{H}_\mathcal{X}$ that is parameterized by the value of $Z$. Given a value $z$ of $Z$, it maps $x \in \mathcal{X}$ to $\mu_{P_{X|Z=z}}(x) := \mathbb{E}_{P_{X|Z=z}}[k_\mathcal{X}(X, x)]$. Then, HSCIC is a mapping from a value of $Z$ to a non-negative real defined as follows: for all $z \in \mathcal{Z}$,

$$\text{HSCIC}(X, Y | Z = z) := \|\mu_{P_{XY|Z=z}} - \mu_{P_{X|Z=z}} \otimes \mu_{P_{Y|Z=z}}\|^2_{\mathcal{H}_\mathcal{X} \otimes \mathcal{H}_\mathcal{Y}}$$

When $k_\mathcal{X} \otimes k_\mathcal{Y}$ satisfies the condition mentioned from above, $X \perp\!\!\!\perp Y \mid Z$ if and only if $\mathbb{E}_{P_Z}[\text{HSCIC}(X, Y|Z)] = 0$ [30]. We use an empirical estimator of the expectation here, denoted by $\widehat{\text{HSCIC}}(X, Y|Z)$. See Section A in the supplementary material for more details for definition of HSCIC and its empirical estimator.

## 3.2    Imitation learning from observation histories

We consider a Partially Observable Markov Decision Process (POMDP) [19] without reward, which is defined as a tuple of $\langle \mathcal{S}, \mathcal{Z}, \mathcal{A}, O, P, \rho_0 \rangle$. Here $\mathcal{S}$, $\mathcal{Z}$ and $\mathcal{A}$ are an (underlying) state space, an observation space and an action space, respectively. The next $O : \mathcal{S} \to \Delta\mathcal{Z}$ specifies the conditional probability $O(z|s)$ of an observation $z \in \mathcal{Z}$ given a state $s \in \mathcal{S}$. Finally, $\rho_0 \in \Delta S$ defines the probability $\rho_0(s_0)$ that the process starts from $s_0 \in \mathcal{S}$, and $P : \mathcal{S} \times \mathcal{A} \to \Delta\mathcal{S}$ defines the probability $P(s'|s, a)$ of transitioning to state $s'$ when action $a \in \mathcal{A}$ is performed in state $s \in \mathcal{S}$.

Assume a (stochastic) expert policy $\pi^E : \mathcal{S} \to \Delta\mathcal{A}$ that is defined as a conditional probability $\pi^E(a|s)$ of the expert's performing an action $a \in \mathcal{A}$ given a state $s \in \mathcal{S}$. Also, assume that we have a dataset $\mathcal{D} = \{\tau^{(1)}, \ldots, \tau^{(N)}\}$ of the expert trajectories where each $\tau = \{(z_t, a_t)\}_{t=0}^T$ is sampled by

$$s_0 \sim \rho_0,\ a_t^E \sim \pi^E(\cdot|s_t),\ z_t \sim O(\cdot|s_t),\ s_{t+1} \sim P(\cdot|s_t, a_t)\ \text{ for } t \in \{0, 1, ..., T\}.$$

That is, $\tau$ in the dataset is drawn from the following joint distribution of all observations and actions:

$$\tau \sim p_D(z_{0:T}, a_{0:T}^E) = \int \rho_0(s_0) \prod_{t=0}^T O(z_t|s_t)\pi^E(a_t^E|s_t)P(s_{t+1}|s_t, a_t^E)ds_{0:T+1}.$$

In our study, we consider POMDP scenarios where an ideal imitator policy is able to match the expert policy's performance, even when the imitator's actions are solely determined by observation histories. Specifically, we focus on situations where the observation histories $z_{t-w-1:t}$ (for some fixed $1 \le w \le T - 1$) encompass all information about the true states $s_t$ utilized by the expert policy.

In this setting, our goal is to learn an imitator policy $\pi^I : \mathcal{Z}^w \to \Delta\mathcal{A}$ that acts as closely to $\pi^E$ as possible on the given dataset $\mathcal{D}$. To achieve this goal, we consider the following joint distribution of actions of both $\pi^I$ and $\pi^E$:

$$p(a_{0:T}^I, a_{0:T}^E) =$$
$$\int \rho_0(s_0) \prod_{t=0}^T O(z_t|s_t)\pi^I(a_t^I|z_{t-w+1:t})\pi^E(a_t^E|s_t)P(s_{t+1}|s_t, a_t^E)ds_{0:T+1}dz_{0:T+1}. \quad (1)$$

## 4  Behavior Cloning with Past Action Leakage Regularization

In this section, we propose a framework that effectively mitigates the past action leakage problem in IL. We first define the problem by formalizing the absence of leaked past-action information via conditional independence (Section 4.1). Then, we compare several regularization-based approaches that attempt to achieve the absence of such information in the context of offline IL (Section 4.2). These approaches performs regularized BC where the regularizer is derived from a metric for measuring the amount of conditional dependence among random variables. The choice of the metric differentiates these approaches, and our comparison advocates the use of the HSCIC-based approach.

### 4.1  Past action leakage problem in imitation learning

When a policy takes an observation history as an input, the input history may include the information about past actions unexpectedly. The inclusion of such information can have detrimental effects in the context of offline imitation learning by confusing the imitator and making it fail to predict expert actions accurately. Intuitively, the past action leakage problem in imitation learning refers to this failure of the imitator due to the leakage of such harmful past action information.

To express this intuition formally, for each timestep $t$, let $A_t^E$ and $A_t^I$ denote random variables of expert action and imitator action at timestep $t$. Note that for each $0 < t \le T$, the joint distribution of the three random variables $A_{t-1}^E$, $A_t^E$ and $A_t^I$ is

$$p(a_{t-1}^E, a_t^E, a_t^I) = \int p(a_{0:T}^I, a_{0:T}^E)da_{0:t-2}^E da_{0:t-1}^I da_{t+1:T}^E da_{t+1:T}^I$$

where $p(a_{0:T}^I, a_{0:T}^E)$ is the distribution in Eq. (1).

We formalize the absence of the leakage of harmful past-action information by conditional independence between the imitator's current actions and the expert's previous actions:

$$A_t^I \perp\!\!\!\perp A_{t-1}^E \mid A_t^E \quad \text{for all } 0 < t \le T. \quad (2)$$

This conditional independence says that the imitator's current action never depends on some information that is only about the expert's past action but not about the expert's current action. This past-specific information corresponds to harmful information in our intuitive explanation from above.

Ideally we would like to achieve conditional independence in Eq. (2), which ensures the absence of leaked past-action information that was harmful to the imitator. However, in practice, we can achieve it only approximately, so that we need a quantitative measure for conditional independence or the lack of conditional independence between $A_{t-1}^E$ and $A_t^E$ given $A_t^I$. Such a measure is also needed to design an offline IL algorithm that does not suffer from such leaked harmful past-action information. In the following subsection, we consider two quantitative measures for the lack of conditional independence, namely, (1) conditional mutual information (CMI) and (2) Hilbert-Schmidt Conditional Independence Criterion (HSCIC).

### 4.2  Behavior cloning with past action leakage regularization

We aim to learn the representation $\varphi_t$ of observation history $z_{t-w+1:t}$ that removes any unnecessary information on the past action. To simplify the notation, let $t_w$ denote $t - w + 1$. Our method is based on the following observation.

**Theorem 1.** *Let $A_t^I$ be the action from the imitator policy $\pi^I(a_t|\varphi_t)$ based on the representation $\varphi_t$ of observation history. Then, $\varphi_t \perp\!\!\!\perp A_{t-1}^E \mid A_t^E \implies A_t^I \perp\!\!\!\perp A_{t-1}^E \mid A_t^E$.*

*Proof.* See Section B in the supplementary material. □

To this end, we formulate our objective of regularized BC framework as follows:

$$\mathcal{L}(\pi, \varphi; \mathcal{D}, \alpha) := \mathcal{L}_{\mathrm{bc}}(\pi, \varphi; \mathcal{D}) + \alpha \cdot \mathcal{L}_{\mathrm{reg}}(\varphi; \mathcal{D}), \tag{3}$$

where $\mathcal{L}_{\mathrm{bc}}$ is an BC objective such that $\mathcal{L}_{\mathrm{bc}}(\pi, \varphi; \mathcal{D}) := \mathbb{E}_{(z_{t_w:t}, a_t^E) \sim \mathcal{D}, \varphi_t \sim \varphi(z_{t_w:t})} \left[ -\log \pi(a_t^E | \varphi_t) \right]$ and $\mathcal{L}_{\mathrm{reg}}$ is a past action leakage regularization objective. For notational simplicity, we abbreviate the expectation with respect to $(a_{t-1}^E, z_{t_w:t}, a_t^E) \sim \mathcal{D}, \varphi_t \sim \varphi(z_{t_w:t})$ to $\mathbb{E}_{\varphi_t}$. In the following subsections, we discuss candidates for $\mathcal{L}_{\mathrm{reg}}$.

### 4.2.1 Information-theoretic regularization

A straightforward way to address the conditional independence we discussed in Section 4.1 is minimizing the CMI, which can be decomposed by its definition into two conditional entropy terms:

$$I(a_{t-1}^E; \varphi_t \mid a_t^E) = H(a_{t-1}^E \mid a_t^E) - H(a_{t-1}^E \mid \varphi_t, a_t^E).$$

It is important to note that $H(a_{t-1}^E \mid a_t^E)$ is determined by the data distribution $\mathcal{D}$ and thus constant. As a result, we can simply consider the minimization of $-H(a_{t-1}^E \mid \varphi_t, a_t^E)$.

$$\mathcal{L}_{\mathrm{reg-Ent}}(\varphi_t; a_{t-1}^E, a_t^E) := -H(a_{t-1}^E \mid \varphi_t, a_t^E) = \mathbb{E}\left[\log p(a_{t-1}^E \mid \varphi_t, a_t^E)\right]. \tag{4}$$

Interestingly, Eq. (4) coincides with the negative conditional entropy maximization objective in FCA [44]. Since the direct computation of Eq. (4) requires to know the intractable distribution $p(a_{t-1}^E \mid \varphi_t, a_t^E)$, FCA trains an additional prediction model $\hat{p}(\hat{a}_{t-1} \mid \varphi_t, a_t)$ to estimate the negative entropy with $-\hat{H}(a_{t-1}^E \mid \varphi_t, a_t^E) := \mathbb{E}_{\varphi_t}[\log \hat{p}(a_{t-1}^E \mid \varphi_t, a_t^E)]$. However, this approach faces the following challenges:

1. The estimated negative entropy $-\hat{H}$ lower bounds Eq. (4), while an upper bound would be desirable for minimizing the objective. Consequently, it imposes nested (minimax) optimization; minimizing with respect to $\varphi_t$ after maximizing with respect to $\hat{p}$.

2. It is required to train an additional neural network to model $\hat{p}$, which consumes additional computational cost.

3. Assuming that the family of variational distribution $\hat{p}$ to be Gaussian, FCA tries to tighten the lower bound estimation $-\hat{H}$ with respect to $\hat{p}$ by minimizing reconstruction error of $a_{t-1}^E$. The assumption restricts the flexibility of $\hat{p}$ and thus the estimation can be inaccurate.

To avoid inaccurate estimation of the entropy, we can also decompose CMI into two MI terms by the chain rule of MI as an alternative:

$$I(a_{t-1}^E; \varphi_t \mid a_t^E) = I(a_{t-1}^E; \varphi_t, a_t^E) - I(a_{t-1}^E; a_t^E),$$

Similar to $H(a_{t-1}^E \mid a_t^E)$, $I(a_{t-1}^E; a_t^E)$ can be safely ignored in the regularization. As a result, the regularization is about simply minimizing $I(a_{t-1}^E; \varphi_t, a_t^E)$ while ignoring the constant MI term.

$$\mathcal{L}_{\mathrm{reg-MI}}(\varphi_t; a_{t-1}^E, a_t^E) := I(a_{t-1}^E; \varphi_t, a_t^E). \tag{5}$$

Since the direct computation of Eq. (5) requires to know densities of $a_{t-1}^E, \varphi_t, a_t^E$, one needs to train sample-based MI estimators [3, 16, 28, 34]. Thanks to those estimators, the estimated MI can be minimized without confining any distribution (to be Gaussian). However, this approach still has issues similar to FCA such that (1) it lower bounds MI and consequently imposes (nested) minimax optimization and (2) it introduces an additional neural network particularly sensitive to hyperparameters.

#### 4.2.2 HSCIC regularization

To bypass those shortcomings in information-theoretic regularization, we consider HSCIC [30], a kernel-based conditional independence metric for the past action leakage regularization.

$$\mathcal{L}_{\mathrm{reg-HSCIC}}(\varphi_t; a_{t-1}^E, a_t^E) := \mathrm{HSCIC}(\varphi_t, a_{t-1}^E | a_t^E). \tag{6}$$

Let $\mathcal{H}_\mathcal{A}, \mathcal{H}_\Phi$ be RKHSs over $\mathcal{A}, \Phi$ and $k_\mathcal{A}, k_\Phi$ be their associated kernels, where $\Phi$ is the representation space induced by the encoder $\varphi$. Given $n$ samples $\{(\varphi_t(i), a_{t-1}^E(i), a_t^E(i))\}_{i=1}^n$ from $\mathcal{D}$ and $\varphi$, let $\mathbf{K}_{\varphi_t}, \mathbf{K}_{a_{t-1}^E}, \mathbf{K}_{a_t^E}$ be the $n \times n$ kernel matrices where $[\mathbf{K}_{\varphi_t}]_{i,j} = k_\Phi(\varphi_t(i), \varphi_t(j)), [\mathbf{K}_{a_{t-1}^E}]_{i,j} = k_\mathcal{A}(a_{t-1}^E(i), a_{t-1}^E(j)), [\mathbf{K}_{a_t^E}]_{i,j} = k_\mathcal{A}(a_t^E(i), a_t^E(j))$. Then, HSCIC estimator for past action leakage regularization can be defined as follows:

$$
\begin{aligned}
\widehat{\mathrm{HSCIC}}(\varphi_t, a_{t-1}^E | a_t^E) := \frac{1}{n}\mathrm{trace}\Big( &\mathbf{K}_{a_t^E}^\top \mathbf{W}(\mathbf{K}_{\varphi_t} \odot \mathbf{K}_{a_{t-1}^E})\mathbf{W}^\top \mathbf{K}_{a_t^E} \\
&- 2\mathbf{K}_{a_t^E}^\top \mathbf{W}(\mathbf{K}_{\varphi_t}\mathbf{W}^\top \mathbf{K}_{a_t^E} \odot \mathbf{K}_{a_{t-1}^E}\mathbf{W}^\top \mathbf{K}_{a_t^E}) \\
&+ (\mathbf{K}_{a_t^E}^\top \mathbf{W}\mathbf{K}_{\varphi_t}\mathbf{W}^\top \mathbf{K}_{a_t^E}) \odot (\mathbf{K}_{a_t^E}^\top \mathbf{W}\mathbf{K}_{a_{t-1}^E}\mathbf{W}^\top \mathbf{K}_{a_t^E})\Big)
\end{aligned} \tag{7}
$$

where $\mathbf{W} = (\mathbf{K}_{a_t^E} + n\lambda\mathbf{I})^{-1}$, $\lambda > 0$ is a ridge regression coefficient, $\odot$ is the element-wise matrix multiplication.

Since Eq. (7) can be estimated using the samples from the joint distribution $p(\varphi_t, a_{t-1}^E, a_t^E)$, we can directly plug-in HSCIC estimates into the regularization objective. By leveraging this estimator, HSCIC regularization offers several advantages compared to the conditional entropy regularization and MI regularization objectives discussed earlier:

1. Direct computation of the closed-form solution in Eq. (7) allows HSCIC regularization to bypass any nested optimization.

2. HSCIC regularization does not employ any additional deep neural networks that require careful hyperparameters.

3. Since HSCIC is a non-parametric measure, it does not impose any parametric assumption on the data distribution and does not require any density estimation.

These advantages strongly imply that promoting conditional independence via the HSCIC estimator will be more desirable compared to other estimators, thereby improving the overall effectiveness of the regularization. Hence, we propose HSCIC regularization to address the past action leakage problem, which we call PALR.

## 5 Experiment

In this section, we present the experimental results of our approach. Initially, we investigate the correlation between the extent of past action information leakage and BC's performance. Subsequently, we compare our approach with several offline ILOH baseline methods across four continuous control tasks from the MuJoCo simulator [41]: hopper, walker2d, halfcheetah, and ant, as well as one pixel-based autonomous driving task from the CARLA simulator [9]: carla-lane [2].

**Dataset**  For tasks from the MuJoCo simulator, we transformed them into POMDP scenarios for ILOH by excluding specific state variables (such as velocity information). The remaining state variables, like positional information and joint angles, were treated as observation variables at individual timesteps[3]. We organized these observations into fixed-size stacks to configure each

---

[2]The D4RL benchmark also featured another task, carla-town, but none of the algorithms surpassed the random policy's performance, indicating a lack of meaningful imitation results. Further details can be found in Section D.2.

[3]Details about the composition of observation variables for each task are available in Section C.1.

problem setting, denoted as `[envname]-W[stacksize]` with stack sizes $w \in \{2, 4\}$. For a task from the CARLA simulator, we used pixel images as observations for ILOH. To extract features from these observations, we employed a pretrained ResNet [15], keeping its parameters fixed during training. In `carla-lane` task, we stacked the extracted features with a fixed stack size of $w = 3$ and used it as input for the policy. All our experiments utilized expert demonstrations from the D4RL benchmark dataset [10] to ensure the validity and reliability of our results.

**Evaluation Metric**    For the performance evaluation of the learned policy, we measure the normalized score that ranges from 0 to 100. To evaluate how much the past action information is leaked into a policy $\pi$, we measure $\widehat{\text{HSCIC}}(a_t^I; a_{t-1}^E | a_t^E)$ using the estimator (7) in the held-out dataset. To estimate HSCIC estimator, we fix a ridge regression coefficient $\lambda = 10^{-5}$ and all kernels are chosen as Gaussian kernels with the bandwidth $\sigma^2 = 1$.

### 5.1 Relationship between past action information leakage and performance

To see if the problem of the past action information leakage occurs, we conduct an empirical study using BC from observation histories with stack size $w = 2$. The objective of this experiment is to confirm the correlation between the performance and the degree of past action information leakage, similar to FCA [44].

**Experimental setup**    To achieve multiple policies at different levels, we train BC policies for 500K steps with 5 different seeds and 7 different training dataset size $N \in \{100\text{K}, 50\text{K}, 30\text{K}, 10\text{K}, 5\text{K}, 3\text{K}, 1\text{K}\}$ (the number of observation history-action pairs). Using a held-out dataset composed of 2,000 samples, the degree of past action information leakage is measured by $\widehat{\text{HSCIC}}(a_t^I; a_{t-1}^E | a_t^E)$ of each policy $\pi^I$.

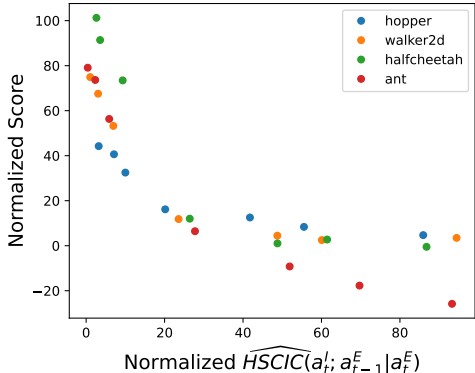

Figure 1: Negative correlation between HSCIC estimates and performance of BC with 7 different dataset sizes.

**Results**    To compare $\widehat{\text{HSCIC}}$ of fully trained BC in all tasks, we normalize $\widehat{\text{HSCIC}}$ of each policy using the maximum and minimum values in each task. Grouping trained policies by the dataset size, we report the mean value of both the normalized score and the normalized HSCIC estimates in Figure 1. It shows that there are clear negative correlations between the normalized HSCIC and the normalized score in all environments. The result implies that BC from observation histories tends to train a policy in which there is a negative correlation between the conditional dependence related to its action and the performance. This insight suggests a potential opportunity for employing regularization methods that enforce conditional independence.

### 5.2 Performance Evaluation

**Experimental setup**    We evaluate the performance of our method and offline baseline methods across 5 environments see effectiveness of our method. We train policies using 5 different algorithms, including our method, for 500K, 1M training steps MuJoCo and CARLA tasks respectively with 5 different seeds.

**Baseline methods**    We compare our method with 6 offline ILOH baseline methods: BC, KF [45], PrimeNet [46] , RAP [6] [4], FCA [44], MINE [3]. While all of these are commonly based on BC, we specify their differences as follows:

---

[4]For the sake of simplicity, here we refer to the baselines [6, 45] as KF (Keyframe-Focused visual imitation learning), RAP (Residual Action Prediction) respectively, following the titles of their respective papers.

| Task | w | BC | KF | PrimeNet | RAP | FCA | MINE | PALR (Ours) |
|---|---|---|---|---|---|---|---|---|
| hopper | 2 | $32.5 \pm 2.9$ | $32.0 \pm 1.9$ | $30.0 \pm 1.6$ | $20.2 \pm 1.4$ | $31.9 \pm 2.5$ | $25.0 \pm 1.9$ | $\mathbf{42.0 \pm 2.4}$ |
| | 4 | $47.7 \pm 3.4$ | $45.7 \pm 1.0$ | $45.3 \pm 2.8$ | $32.6 \pm 2.6$ | $36.9 \pm 2.4$ | $37.6 \pm 3.1$ | $\mathbf{58.4 \pm 2.8}$ |
| walker2d | 2 | $53.0 \pm 2.7$ | $50.0 \pm 2.3$ | $48.5 \pm 3.3$ | $15.8 \pm 2.0$ | $63.1 \pm 2.7$ | $58.6 \pm 5.5$ | $\mathbf{79.8 \pm 2.3}$ |
| | 4 | $63.2 \pm 6.3$ | $77.4 \pm 2.0$ | $79.2 \pm 3.3$ | $25.4 \pm 2.1$ | $\mathbf{81.9 \pm 3.3}$ | $68.7 \pm 6.7$ | $\mathbf{83.4 \pm 5.4}$ |
| halfcheetah | 2 | $74.1 \pm 2.3$ | $64.3 \pm 1.4$ | $61.5 \pm 1.9$ | $63.9 \pm 2.1$ | $78.2 \pm 2.8$ | $76.3 \pm 1.9$ | $\mathbf{86.4 \pm 1.1}$ |
| | 4 | $68.4 \pm 2.6$ | $55.7 \pm 4.1$ | $45.5 \pm 1.7$ | $59.0 \pm 2.7$ | $69.9 \pm 2.6$ | $73.4 \pm 2.4$ | $\mathbf{79.1 \pm 4.3}$ |
| ant | 2 | $56.3 \pm 3.5$ | $54.9 \pm 1.7$ | $51.7 \pm 2.4$ | $44.1 \pm 1.2$ | $51.1 \pm 2.2$ | $53.9 \pm 1.9$ | $\mathbf{59.6 \pm 3.0}$ |
| | 4 | $\mathbf{64.4 \pm 1.8}$ | $48.6 \pm 3.8$ | $58.2 \pm 1.9$ | $48.6 \pm 2.6$ | $57.7 \pm 1.3$ | $56.6 \pm 1.8$ | $\mathbf{64.6 \pm 2.5}$ |
| carla-lane | 3 | $52.5 \pm 6.2$ | $66.6 \pm 2.1$ | $58.2 \pm 2.2$ | $25.3 \pm 5.4$ | $57.1 \pm 3.1$ | $60.1 \pm 4.1$ | $\mathbf{72.9 \pm 2.6}$ |

Table 1: Performance evaluation of baseline and regularization methods. The normalized scores averaged over the final 50 evaluations during training and we report mean and standard error over 5 different seeds. The rightmost three algorithms are incorporated into our regularization framework. The method with the highest mean score and its competitive methods (within standard error) are highlighted in bold in each problem setting.

- **BC** : the standard BC algorithm from observation history, which optimizes an objective without any regularization term ($\alpha = 0$).

- **KF** [45] : the weighted BC algorithm that assigns higher weights to keyframes, which contain actions that are highly predictable from their corresponding action histories.

- **PrimeNet** [46] : a supervised learning method designed to prevent undesirable shortcuts by leveraging additional key inputs.

- **RAP** [6] : it employs a dual-stream of policy representation that learns from both observation history and individual observations, maximizing a lower bound that enforces conditional dependence between the representation and expert action, given the past action.

- **FCA** [44] : it maximizes the conditional entropy that corresponds to Eq. (4) with adversarial training, which is an instance of our regularized BC framework.

- **MINE** [3] : it minimizes the MI estimate corresponding to Eq. (5) using MINE estimator [3], which is one of the representative sample-based neural estimators for MI. It is also an instance of our regularized BC framework.

Our method regularizes HSCIC with its estimator defined as Eq. (7), where Gaussian kernel with fixed bandwidth $\sigma^2 = 1$ are used for all kernels of $\varphi_t, a_{t-1}^E, a_t^E$ and a ridge regression coefficient $\lambda = 10^{-5}$. Across all regularization methods, we searched for the optimal $\alpha$ according to the best mean normalized score. Further implementation details can be found in Section C.

**Results** Table 1 summarizes the results of performance evaluation for each problem settings. Our method significantly outperforms other baselines in 7 settings out of 9 and shows competitive performance in the rest 2 settings. In particular, our method shows strong performance in `carla-lane`, highlighting its effectiveness in enhancing performance within high-dimensional offline ILOH scenarios. We observe that RAP shows the least competitive performance across all tasks. This is because they do not have any penalization of the dependence between the imitator action and expert action in their objective function. However, we also observe that FCA and MINE fail to show consistent improvement over BC in most tasks except `walker2d`. This is because the lower bound estimators of their regularization objectives are not sufficiently accurate even at the cost of their inefficient alternating optimization. On the other hand, our method consistently outperforms BC, which clearly indicates the effect of promoting the conditional independence as we discussed in Section 4.1.

To better understand the degenerate performance of FCA and MINE, we evaluate $\widehat{\text{HSCIC}}(a_t^I; a_{t-1}^E | a_t^E)$ of each regularization method during training using the held-out dataset in Figure 2a. In contrast to our method, the result clearly demonstrates that FCA and MINE commonly

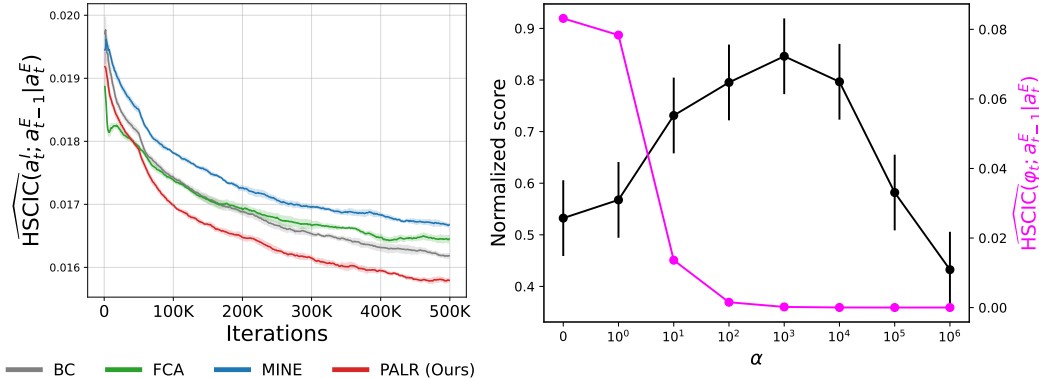

(a) HSCIC estimate during training on `hopper-W4`.    (b) Varying regularization coefficient in `walker2d-W2`.

Figure 2: Investigating past action leakage regularization methods on D4RL dataset.

fail to promote conditional independence throughout the learning process, elucidating their underperformance. To further investigate, we measure the HSCIC and conditional MI estimates of the finalized imitator policies across 8 MuJoCo settings (refer to Section D.5). In essence, our method adeptly mitigates the leakage of past action information, surpassing the efficacy of alternative approaches.

**Regularization coefficient**    In this experiment, we aim to ascertain how the selection of coefficient $\alpha$ influences the performance of our method. We evaluate the asymptotic performance of our method with varying values of $\alpha \in \{0, 10^0, 10^1, 10^2, 10^3, 10^4, 10^5, 10^6\}$ on the `walker2d-W2` problem setting, which shows the largest performance gap between PALR and BC among our problem settings. As depicted in Figure 2b, the converged regularization loss $\mathcal{L}_{\mathrm{reg-HSCIC}}$ progressively decreases as $\alpha$ increases. Notably, the $\alpha$ that minimizes the regularization loss is not necessarily equal to the optimal hyperparameter that maximizes performance. We observe that the selection of $\alpha$ is important, as it adjusts the trade-off between robustness to the leakage and alignment with expert data.

## 6    Conclusion and Future Work

Grounded in the classical notion of conditional independence, we proposed a principled regularization framework for BC that mitigates past action information leakage problem. Within our framework, we have explored multiple choices of conditional independence metric and compared their estimators. Finally, we identified that our method with HSCIC estimator is the most favorable regularization of BC over other choices in terms of robustness to the leaked information of past action. In an extensive set of experiments on D4RL datasets, we empirically showed that our method significantly outperforms baseline methods. We also observed in our experiments that all the comparing methods including ours were sensitive to the choice of $\alpha$. Without assuming any interaction with the environment, it is challenging to find the optimal $\alpha$ only with the given offline dataset. We believe that the discovery of optimal $\alpha$ in offline manner would be an interesting research topic, which we leave as future work.

## Acknowledgments and Disclosure of Funding

This work was partly supported by IITP grant funded by MSIT (No.2020-0-00940, Foundations of Safe Reinforcement Learning and Its Applications to Natural Language Processing; No.2022-0-00311, Development of Goal-Oriented Reinforcement Learning Techniques for Contact-Rich Robotic Manipulation of Everyday Objects; No.2019-0-00075, AI Graduate School Program (KAIST); No.2021-0-02068, AI Innovation Hub), NRF of Korea (NRF2019R1A2C1087634; RS-2023-00279680; NRF-2018R1A5A1059921), Field-oriented Technology Development Project for Customs Administration through NRF of Korea funded by the MSIT and Korea Customs Service (NRF2021M3I1A1097938), ETRI grant (22ZS1100, Core Technology Research for Self-Improving Integrated AI System), KAIST-NAVER Hypercreative AI Center.

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

# Supplementary Material

## Contents

# A  Details of HSCIC and Its Empirical Estimation

## A.1  Definitions

**Definition 1.** *(Reproducing kernel Hilbert space)*
*Let $\mathcal{H}_{\mathcal{X}}$ be a Hilbert space of functions $f : \mathcal{X} \to \mathbb{R}$ functions which the metric is defined by inner product $\langle \cdot, \cdot \rangle_{\mathcal{H}_{\mathcal{X}}}$. A symmetric function $k_{\mathcal{X}} : \mathcal{X} \times \mathcal{X} \to \mathbb{R}$, which is called a reproducing kernel of $\mathcal{H}_{\mathcal{X}}$, that satisfies*

 1. $\forall x \in \mathcal{X}$ $\qquad\qquad k_{\mathcal{X}}(x, \cdot) \in \mathcal{H}_{\mathcal{X}}$

 2. $\forall x \in \mathcal{X}, \forall f \in \mathcal{H}_{\mathcal{X}}$ $\quad \langle f, k_{\mathcal{X}}(x, \cdot) \rangle_{\mathcal{H}_{\mathcal{X}}} = f(x)$ *(called **reproducing property**)*

*A space $\mathcal{H}_{\mathcal{X}}$ is called a **reproducing kernel Hilbert spaces (RKHS)** corresponding a reproducing kernel $k_{\mathcal{X}}$.*

**Definition 2.** *(Kernel Mean Embedding)*
*Given a distribution $P_X$ on $\mathcal{X}$, assume $\int_{\mathcal{X}} \sqrt{k_{\mathcal{X}}(x,x)} dP_X(x) < \infty$. We define **the kernel mean embedding** $\mu_{P_X}$ of $P_X$ as $\mu_{P_X}(\cdot) = \int_{\mathcal{X}} k_{\mathcal{X}}(x, \cdot) dP_X(x)$.*

**Definition 3.** *(Maximum Mean Discrepancy)*
*Given distributions $P, Q$ on $\mathcal{X}$ and suppose kernel mean embeddings of $P, Q$ exist on $\mathcal{H}$, denote $\mu_P, \mu_Q$. Then, the **maximum mean discrepancy (MMD)** between $P_X, Q_Y$ is defined as follows:*

$$\mathrm{MMD}(P, Q; \mathcal{H}) = \|\mu_P - \mu_Q\|_{\mathcal{H}}$$

**Definition 4.** *(Hilbert-Schmidt Norm and Operator)*
*Let $\mathcal{F}, \mathcal{G}$ be Hilbert spaces. Define $(f_i)_{i \in I}, (g_j)_{j \in J}$ to be orthonormal basis for $\mathcal{F}, \mathcal{G}$ respectively. Define two linear operators $L : \mathcal{G} \to \mathcal{F}, M : \mathcal{G} \to \mathcal{F}$.*

 1. *The **Hilbert-Schmidt norm** of the operator $L$ is defined as:*

$$\|L\|_{\mathrm{HS}}^2 = \sum_{j \in J} \|Lg_j\|_{\mathcal{F}}^2$$
$$= \sum_{j \in J} \sum_{i \in I} |\langle Lg_j, f_i \rangle_{\mathcal{F}}|^2$$

 2. *The operator $L$ is called **Hilbert-Schmidt Operator** when $\|L\|_{\mathrm{HS}}^2$ is finite.*

**Definition 5.** *(Cross-covariance Operator)*
*Given RKHS $\mathcal{F}, \mathcal{G}$ with kernels $k, l$, feature maps $\phi, \psi$ respectively. Define the uncentered covariance operator $\tilde{C}_{XY} : \mathcal{G} \to \mathcal{F}$ and the **cross-covariance operator** $C_{XY} : \mathcal{G} \to \mathcal{F}$ as follows:*

$$\langle \tilde{C}_{XY}, A \rangle_{HS} = \mathbb{E}_{x,y} \langle \phi(x) \otimes \psi(y), A \rangle_{\mathrm{HS}}$$
$$C_{XY} := \tilde{C}_{XY} - \mu_X \otimes \mu_Y$$

**Definition 6.** *(Hilbert-Schmidt Independence Criterion)*
*The **Hilbert-Schmidt Independence Criterion (HSIC)** is defined as Hilbert-Schmidt norm of the cross-covariance operator $C_{XY}$, i.e., $\mathrm{HSIC}(P_{XY}; \mathcal{F}, \mathcal{G}) := \|C_{XY}\|_{\mathrm{HS}}^2$. Equivalently, we can define HSIC as MMD between $P_{XY}$ and $P_X P_Y$ with the product kernel, i.e.*

$$\mathrm{HSIC}(P_{XY}; \mathcal{F}, \mathcal{G}) = \mathrm{MMD}^2(P_{XY}, P_X P_Y; \mathcal{H}_{\kappa}),$$

*where $\kappa((x, y), (x', y')) = k(x, x')l(y, y')$.*

**Definition 7.** *(Conditional Mean Embedding)*
*Assuming a random variable $X$ satisfies $\int_{\mathcal{X}} \sqrt{k(x,x)} dP < \infty$. Define the **conditional mean embedding** of $X$ given $Z$ as $\mu_{P_{X|Z}}(\cdot) := \mathbb{E}_{X|Z}[k_{\mathcal{X}}(X, \cdot)|Z]$.*

Note that $\mu_{P_{X|Z}}$ is a random variable. In the main text of our paper, we denote the realization of $\mu_{P_{X|Z}}$ with $Z = z$ as $\mu_{P_{X|Z=z}}$.

**Definition 8.** *(Hilbert-Schmidt Conditional Independence Criterion)*
*Define the **Hilbert-Schmidt Conditional Independence Criterion (HSCIC)** between $X$ and $Y$ given $Z$ to be*
$$\mathrm{HSCIC}(X, Y|Z) = \|\mu_{P_{XY|Z}} - \mu_{P_{X|Z}} \otimes \mu_{P_{Y|Z}}\|^2_{\mathcal{H}_\mathcal{X} \otimes \mathcal{H}_\mathcal{Y}}.$$

## A.2 Empirical estimation

We start to explain from decoupling the conditional mean embedding into their deterministic function and conditional random variable.

**Lemma 1.** *(Decomposition of conditional mean embedding, Theorem 4.1 of [30])*
*There exists a deterministic function $F_{P_{X|Z}} : \mathcal{Z} \to \mathcal{H}_\mathcal{X}$, which satisfies $\mu_{P_{X|Z}} = F_{P_{X|Z}} \circ Z$.*

By this lemma, we can decompose conditional mean embeddings and HSCIC into some deterministic function and conditioned random variable $Z$, i.e., $\mu_{P_{X|Z}} = F_{P_{X|Z}} \circ Z, \mu_{P_{Y|Z}} = F_{P_{Y|Z}} \circ Z, \mu_{P_{XY|Z}} = F_{P_{XY|Z}} \circ Z, \mathrm{HSCIC}(X, Y|Z) = F_{\mathrm{HSCIC}_{X,Y|Z}} \circ Z$ respectively. By using vector-valued RKHS regression with a regularization parameter $\lambda > 0$ and the representer theorem [23], we can obtain closed-form estimates of $F_{P_{X|Z}}, F_{P_{Y|Z}}, F_{P_{XY|Z}}$ as follows (see [30] for more details):

$$\mathbf{W}_Z := (\mathbf{K}_Z + n\lambda\mathbf{I})^{-1}$$
$$\hat{F}_{P_{X|Z}}(z) = \mathbf{k}_Z^\top(z)\mathbf{W}_Z\mathbf{k}_X(\cdot)$$
$$\hat{F}_{P_{Y|Z}}(z) = \mathbf{k}_Z^\top(z)\mathbf{W}_Z\mathbf{k}_Y(\cdot)$$
$$\hat{F}_{P_{XY|Z}}(z) = \mathbf{k}_Z^\top(z)\mathbf{W}_Z(\mathbf{k}_X(\cdot) \odot \mathbf{k}_Y(\cdot))$$

where $\mathbf{k}_X(\cdot) := (k_\mathcal{X}(x_1, \cdot), ..., k_\mathcal{X}(x_n, \cdot))^\top, \mathbf{k}_Y(\cdot) := (k_\mathcal{Y}(y_1, \cdot), ..., k_\mathcal{Y}(y_n, \cdot))^\top, \mathbf{k}_Z(\cdot) := (k_\mathcal{Z}(z_1, \cdot), ..., k_\mathcal{Z}(z_n, \cdot))^\top, [\mathbf{K}_Z]_{ij} := k_\mathcal{Z}(z_i, z_j)$ and $\odot$ is the element-wise multiplication operator of matrices. By plugging-in conditional mean embedding estimators, we can obtain a closed-form estimator of $F_{\mathrm{HSCIC}_{X,Y|Z}}$ as follows:

$$\hat{F}_{\mathrm{HSCIC}_{X,Y|Z}}(z) := \|\hat{F}_{P_{XY|Z}}(z) - \hat{F}_{P_{X|Z}}(z) \otimes \hat{F}_{P_{Y|Z}}(z)\|^2_{\mathcal{H}_\mathcal{X} \otimes \mathcal{H}_\mathcal{Y}}$$
$$= \mathbf{k}_Z^\top(z)\mathbf{W}_Z(\mathbf{K}_X \odot \mathbf{K}_Y)\mathbf{W}_Z^\top\mathbf{k}_Z(z)$$
$$- 2\mathbf{k}_Z^\top(z)\mathbf{W}_Z(\mathbf{K}_X\mathbf{W}_Z^\top\mathbf{k}_Z(z) \odot \mathbf{K}_Y\mathbf{W}_Z^\top\mathbf{k}_Z(z))$$
$$+ (\mathbf{k}_Z^\top(z)\mathbf{W}_Z\mathbf{K}_X\mathbf{W}_Z^\top\mathbf{k}_Z(z))(\mathbf{k}_Z^\top(z)\mathbf{W}_Z\mathbf{K}_Y\mathbf{W}_Z^\top\mathbf{k}_Z(z))$$
$$\widehat{\mathrm{HSCIC}}(X, Y|Z) = \frac{1}{n}\sum_{i=1}^n \hat{F}_{\mathrm{HSCIC}_{X,Y|Z}}(z_i)$$
$$= \frac{1}{n}\mathrm{trace}\Big(\mathbf{K}_Z^\top\mathbf{W}_Z(\mathbf{K}_X \odot \mathbf{K}_Y)\mathbf{W}_Z^\top\mathbf{K}_Z$$
$$- 2\mathbf{K}_Z^\top\mathbf{W}_Z(\mathbf{K}_X\mathbf{W}_Z^\top\mathbf{K}_Z \odot \mathbf{K}_Y\mathbf{W}_Z^\top\mathbf{K}_Z)$$
$$+ (\mathbf{K}_Z^\top\mathbf{W}_Z\mathbf{K}_X\mathbf{W}_Z^\top\mathbf{K}_Z) \odot (\mathbf{K}_Z^\top\mathbf{W}_Z\mathbf{K}_Y\mathbf{W}_Z^\top\mathbf{K}_Z)\Big) \qquad (8)$$

where $[\mathbf{K}_X]_{ij} := k_\mathcal{X}(x_i, x_j), [\mathbf{K}_Y]_{ij} := k_\mathcal{Y}(y_i, y_j)$.

## B  Proof of Theorem 1

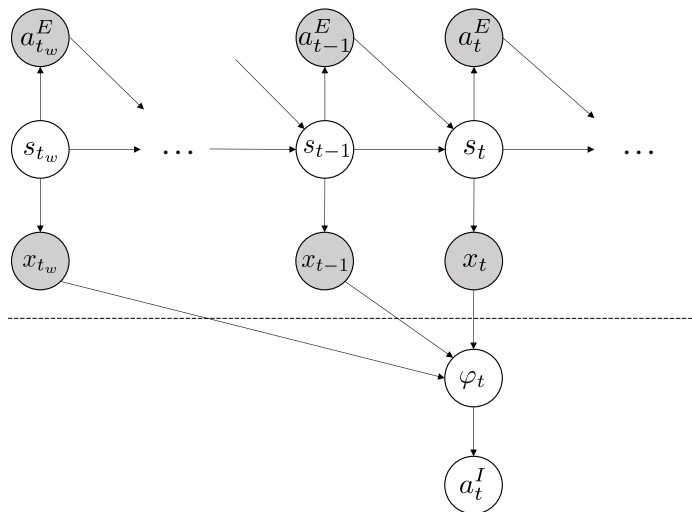

Figure 3: Graphical model of behavior cloning with observation histories.

*Proof.* By the chain rule of MI, following equalities hold:

$$I(\varphi_t; a_{t-1}^E \mid a_t^E) = I(a_{t-1}^E; \varphi_t, a_t^E) - I(a_{t-1}^E; a_t^E)$$
$$I(a_t^I; a_{t-1}^E \mid a_t^E) = I(a_{t-1}^E; a_t^I, a_t^E) - I(a_{t-1}^E; a_t^E)$$

Thus, $I(\varphi_t; a_{t-1}^E \mid a_t^E) \geq I(a_t^I; a_{t-1}^E \mid a_t^E)$ if and only if $I(a_{t-1}^E; \varphi_t, a_t^E) \geq I(a_{t-1}^E; a_t^I, a_t^E)$.

By the chain rule of MI,

$$I(a_{t-1}^E; \varphi_t, a_t^I, a_t^E) = I(a_{t-1}^E; \varphi_t, a_t^E) + I(a_{t-1}^E; a_t^I \mid \varphi_t, a_t^E)^{\nearrow 0}$$
$$= I(a_{t-1}^E; a_t^I, a_t^E) + I(a_{t-1}^E; \varphi_t \mid a_t^I, a_t^E)$$
$$\geq I(a_{t-1}^E; a_t^I, a_t^E)$$

Thus, $I(\varphi_t; a_{t-1}^E \mid a_t^E) \geq I(a_t^I; a_{t-1}^E \mid a_t^E) \ \forall t \geq 0$.  □

Directly regularizing the imitator policy to enforce conditional independence may cause harmful results of training. This is because any policy that ensures the conditional independence does not necessarily yield optimal behavior. For instance, constant actions or actions sampled from uniform random distribution are independent of other random variables and thus satisfy Eq. (2), although they are not optimal in general. Based on Theorem 1, we adjust the representation of policy to inject the conditional independence. The effectiveness of our approach is compared with direct action regularization through an empirical study. The results are provided in Table 9 of Section D.1.

## C  Implementation Details

This section provides detailed specifications and implementation settings.

### C.1  Observation specification

Based on the observation specification from Open AI Gym documentation[5], we take a partial observation in each task by excluding certain variables (e.g. velocity information) in the original state variables. We mark the variables we used in our observation as ◯ and not used are represented as × in the observation column of the corresponding table.

### C.1.1  Hopper

| Index | Description | Unit | Observation |
|:---:|:---:|:---:|:---:|
| 0 | z-coordinate of the top | position (m) | ◯ |
| 1 | angle of the top | angle (rad) | ◯ |
| 2 | angle of the thigh joint | angle (rad) | ◯ |
| 3 | angle of the leg joint | angle (rad) | ◯ |
| 4 | angle of the foot joint | angle (rad) | ◯ |
| 5 | velocity of the x-coordinate of the top | velocity (m/s) | × |
| 6 | velocity of the z-coordinate of the top | velocity (m/s) | × |
| 7 | angular velocity of the angle of the top | angular velocity (rad/s) | × |
| 8 | angular velocity of the thigh hinge | angular velocity (rad/s) | × |
| 9 | angular velocity of the leg hinge | angular velocity (rad/s) | × |
| 10 | angular velocity of the foot hinge | angular velocity (rad/s) | × |

Table 2: Composition of observation variables (`hopper`)

### C.1.2  Walker2d

| Index | Description | Unit | Observation |
|:---:|:---:|:---:|:---:|
| 0 | z-coordinate of the top | position (m) | ◯ |
| 1 | angle of the top | angle (rad) | ◯ |
| 2 | angle of the thigh joint | angle (rad) | ◯ |
| 3 | angle of the leg joint | angle (rad) | ◯ |
| 4 | angle of the foot joint | angle (rad) | ◯ |
| 5 | angle of the left thigh joint | angle (rad) | ◯ |
| 6 | angle of the left leg joint | angle (rad) | ◯ |
| 7 | angle of the left foot joint | angle (rad) | ◯ |
| 8 | velocity of the x-coordinate of the top | velocity (m/s) | × |
| 9 | velocity of the z-coordinate of the top | velocity (m/s) | × |
| 10 | angular velocity of the angle of the top | angular velocity (rad/s) | × |
| 11 | angular velocity of the thigh hinge | angular velocity (rad/s) | × |
| 12 | angular velocity of the leg hinge | angular velocity (rad/s) | × |
| 13 | angular velocity of the foot hinge | angular velocity (rad/s) | × |
| 14 | angular velocity of the thigh hinge | angular velocity (rad/s) | × |
| 15 | angular velocity of the leg hinge | angular velocity (rad/s) | × |
| 16 | angular velocity of the foot hinge | angular velocity (rad/s) | × |

Table 3: Composition of observation variables (`walker2d`)

---

[5]`https://www.gymlibrary.dev/environments/mujoco/`

### C.1.3 HalfCheetah

Open AI Gym documentation does not match with the XML file of `halfcheetah`[6], we denote both descriptions on the following tables.

| Index | Description | Unit | Observation |
|:---:|:---:|:---:|:---:|
| 0 | z-coordinate of the front tip | position (m) | ◯ |
| 1 | angle of the front tip | angle (rad) | ◯ |
| 2 | angle of the second rotor | angle (rad) | ◯ |
| 3 | angle of the second rotor | angle (rad) | ◯ |
| 4 | velocity of the tip along the x-axis | velocity (m/s) | ✕ |
| 5 | velocity of the tip along the y-axis | velocity (m/s) | ✕ |
| 6 | angular velocity of the front tip | angular velocity (rad/s) | ✕ |
| 7 | angular velocity of the second rotor | angular velocity (rad/s) | ✕ |
| 8 | x-coordinate of the front tip | position (m) | ◯ |
| 9 | y-coordinate of the front tip | position (m) | ◯ |
| 10 | angle of the front tip | angle (rad) | ◯ |
| 11 | angle of the second rotor | angle (rad) | ◯ |
| 12 | angle of the second rotor | angle (rad) | ◯ |
| 13 | velocity of the tip along the x-axis | velocity (m/s) | ✕ |
| 14 | velocity of the tip along the y-axis | velocity (m/s) | ✕ |
| 15 | angular velocity of the front tip | angular velocity (rad/s) | ✕ |
| 16 | angular velocity of the second rotor | angular velocity (rad/s) | ✕ |

Table 4: Composition of observation variables (`halfcheetah`) on Gym Documentation.

| Index | Description | Unit | Observation |
|:---:|:---:|:---:|:---:|
| 0 | z-coordinate of the root | position (m) | ◯ |
| 1 | y-coordinate of the root | position (m) | ◯ |
| 2 | angle of the back thigh | angle (rad) | ◯ |
| 3 | angle of the back shin | angle (rad) | ◯ |
| 4 | angle of the back foot | angle (rad) | ✕ |
| 5 | angle of the front thigh | angle (rad) | ✕ |
| 6 | angle of the front shin | angle (rad) | ✕ |
| 7 | angle of the front foot | angle (rad) | ✕ |
| 8 | velocity of the root along the x-axis | velocity (m/s) | ◯ |
| 9 | velocity of the root along the z-axis | velocity (m/s) | ◯ |
| 10 | velocity of the root along the y-axis | velocity (m/s) | ◯ |
| 11 | angular velocity of the back thigh | angular velocity (rad/s) | ◯ |
| 12 | angular velocity of the back shin | angular velocity (rad/s) | ◯ |
| 13 | angular velocity of the back foot | angular velocity (rad/s) | ✕ |
| 14 | angular velocity of the front thigh | angular velocity (rad/s) | ✕ |
| 15 | angular velocity of the front shin | angular velocity (rad/s) | ✕ |
| 16 | angular velocity of the front foot | angular velocity (rad/s) | ✕ |

Table 5: Composition of observation variables (`halfcheetah`) on the XML file.

---

[6]`https://github.com/openai/gym/blob/master/gym/envs/mujoco/assets/half_cheetah.xml`

### C.1.4 Ant

| Num | Description | Unit | Observation |
|---|---|---|---|
| 0 | z-coordinate of the torso (centre) | position (m) | ○ |
| 1 | x-orientation of the torso (centre) | angle (rad) | ○ |
| 2 | y-orientation of the torso (centre) | angle (rad) | ○ |
| 3 | z-orientation of the torso (centre) | angle (rad) | ○ |
| 4 | w-orientation of the torso (centre) | angle (rad) | ○ |
| 5 | angle between torso and first link on front left | angle (rad) | ○ |
| 6 | angle between the two links on the front left | angle (rad) | ○ |
| 7 | angle between torso and first link on front right | angle (rad) | ○ |
| 8 | angle between the two links on the front right | angle (rad) | ○ |
| 9 | angle between torso and first link on back left | angle (rad) | ○ |
| 10 | angle between the two links on the back left | angle (rad) | ○ |
| 11 | angle between torso and first link on back right | angle (rad) | ○ |
| 12 | angle between the two links on the back right | angle (rad) | ○ |
| 13 | x-coordinate velocity of the torso | velocity (m/s) | × |
| 14 | y-coordinate velocity of the torso | velocity (m/s) | × |
| 15 | z-coordinate velocity of the torso | velocity (m/s) | × |
| 16 | x-coordinate angular velocity of the torso | angular velocity (rad/s) | × |
| 17 | y-coordinate angular velocity of the torso | angular velocity (rad/s) | × |
| 18 | z-coordinate angular velocity of the torso | angular velocity (rad/s) | × |
| 19 | angular velocity of angle between torso and front left link | angular velocity (rad/s) | × |
| 20 | angular velocity of the angle between front left links | angular velocity (rad/s) | × |
| 21 | angular velocity of angle between torso and front right link | angular velocity (rad/s) | × |
| 22 | angular velocity of the angle between front right links | angular velocity (rad/s) | × |
| 23 | angular velocity of angle between torso and back left link | angular velocity (rad/s) | × |
| 24 | angular velocity of the angle between back left links | angular velocity (rad/s) | × |
| 25 | angular velocity of angle between torso and back right link | angular velocity (rad/s) | × |
| 26 | angular velocity of the angle between back right links | angular velocity (rad/s) | × |
| 27-110 | contact forces applied to the center of mass of each of the links | force / torque | × |

Table 6: Composition of observation variables (`ant`)

### C.2 Dataset preprocessing

**MuJoCo Tasks** We utilized the expert dataset provided in D4RL benchmark that has a suffix `-expert-v2`. For all experiment, we configured partial observations from the original data based on Section C.1. These observations are standardized with statistics from the entire dataset. Each dataset in D4RL benchmark has a total of 1M transitions. Given the size of the training data $D$, we take the first $D$ samples. Among the last 100K samples in the entire dataset, we also sampled heldout data which are not included in the training data. Given a stacksize $w \in \{2, 4\}$, we configure each transition as (observation history, action, past action) tuple. For the main performance evaluation experiment (Table 1), we use 30K transitions for training.

**CARLA Tasks** We extract features from observation images using the pretrained network ResNet34 [15] and consider these features as policy inputs during training. For the performance evaluation (Table 1, 10), we utilize 1K and 100K transitions for `carla-lane`, `carla-town` respectively.

### C.3 Hyperparameter selection

This section outlines the settings of the hyperparameters for each method. To ensure a fair comparison, we aim to maintain consistent network architecture and policy-related hyperparameters, whenever feasible. For all baselines and our method, we employ a policy architecture consisting of fully-connected layers with [128, 64, 128] hidden units. The layer with 64 units is treated as a policy representation, denoted as $\varphi$.

**KF [45]**  We used a softmax function as a weighting function of the weighted BC in MuJoCo tasks and a step function in CARLA tasks, aligning with the the original paper's experiments. We adopt the action prediction network for the additional network. Additionally, we tuned hyperparameters for the the softmax temperature in MuJoCo tasks, analogous to $\alpha$. In CARLA tasks, we used the same hyperparameters (threshold=0.1, weight=5) consistent with the original paper [45].

**PrimeNet [46]**  We treat observation histories as raw inputs and individual last observations as key inputs, consistent with the approach used in the original experiments. The architecture of the prime network is implemented as the same with the policy.

**RAP [6]**  RAP incorporates two streams of policy representation: (1) one stream derived from observation history (referred to as the memory extraction stream) and (2) the other originating from a single observation (referred to as the policy stream). For the memory extraction stream, we use an architecture of [128, 64, 300] hidden units that outputs predictions of a residual of expert actions. The layer with 64 units is employed as a representation. For the policy stream, we similarly utilize an architecture of [128, 64(+64), 128] which outputs a policy action. The middle units indicate that the policy utilizes both the single observation representation and the memory-extracted representation (with a stop-gradient layer) as inputs.

A summary of hyperparameters is provided in Table 7. The first 3 rows of the table display the hyperparameters universally applied across all methods. The next 4 rows detail hyperparameters for methods that incorporate additional neural networks. The last 2 rows of the table present hyperparameters specific to our method, encompassing hyperparameters for HSCIC estimates.

| Hyperparameter | KF | PrimeNet | RAP | FCA | MINE | PALR (Ours) |
|---|---|---|---|---|---|---|
| Policy distribution | | | Tanh Normal | | | |
| Batch size | | | 1024 | | | |
| BC learning rate | | | 3e-4 | | | |
| Additional network | | | | | | |
|    hidden units | [128,128] | [128,64,128] | [128,64,300] | [300] | [100,100] | - |
|    num of inner updates | 1 | 1 | 1 | 5 | 5 | - |
|    learning rate | 3e-4 | 3e-4 | 3e-4 | 1e-4 | 1e-4 | - |
| IB coefficient | - | - | - | 0.01 | - | - |
| Kernel bandwidth $\sigma^2$ | - | - | - | - | - | 1. |
| Ridge coefficient $\lambda$ | - | - | - | - | - | 1e-5 |

Table 7: The summary of hyperparameters.

**Optimal $\alpha$**  As discussed in Section 5.2, we train 5 policies incorporating different regularization coefficients $\alpha$ within the set $\{0.01, 0.1, 1, 10, 100\}$ for MuJoCo tasks and $\{0.001, 0.1, 10, 1000\}$ for CARLA task. We then select the best $\alpha$ for each problem setting. Table 8 shows the chosen values of $\alpha$ for each method and each problem setting, as utilized in the experimental results of the paper.

| Problem Setting | KF | FCA | MINE | Ours |
|---|---|---|---|---|
| `hopper-W2` | 0.01 | 0.1 | 1 | 10 |
| `hopper-W4` | 0.01 | 0.1 | 0.01 | 10 |
| `walker2d-W2` | 0.01 | 0.1 | 1 | 100 |
| `walker2d-W4` | 0.1 | 0.1 | 1 | 100 |
| `halfcheetah-W2` | 0.01 | 0.1 | 1 | 10 |
| `halfcheetah-W4` | 0.1 | 0.1 | 0.1 | 10 |
| `ant-W2` | 1 | 0.1 | 0.01 | 10 |
| `ant-W4` | 0.01 | 1 | 0.01 | 10 |
| `carla-lane-W3` | - | 0.001 | 10 | 1000 |

Table 8: The summary of the best regularization coefficient $\alpha$.

# D  Additional Experiments

## D.1  Direct regularization on imitator action

To investigate the effectiveness of representation regularization, we compare our method to the one that directly regularizes action in MuJoCo settings. The objective function of the direct method is defined as follows:

$$\mathcal{L}_{\mathrm{PALR-ACT}}(\pi; \mathcal{D}, \alpha) := \mathcal{L}_{\mathrm{bc}}(\pi; \mathcal{D}) + \alpha \cdot \mathbb{E}_{a_t^I \sim \pi(\cdot | z_{t_w:t}), (z_{t_w:t}, a_{t-1}^E, a_t^E) \sim \mathcal{D}} [\widehat{\mathrm{HSCIC}}(a_t^I, a_{t-1}^E | a_t^E)].$$

We optimize the imitator policy $\pi$ with the objective by using the reparameterization trick [21]. The method is also trained with different regularization coefficients $\alpha \in \{0.01, 0.1, 1, 10, 100\}$ and we select the best $\alpha$ of each method. Table 9 summarizes the results. Our representation regularization method (PALR) is more successful in 6 out of the 8 problem settings when compared to direct action regularization. In `ant-W2`, `ant-W4` problem settings, neither method demonstrates significant effectiveness when compared to BC method.

| Task | w | BC | PALR (ours) | PALR-ACT |
|------|---|----|-------------|----------|
| hopper | 2 | $32.47 \pm 2.85$ | $\mathbf{42.01 \pm 2.44}$ | $31.60 \pm 2.81$ |
|  | 4 | $47.65 \pm 3.43$ | $\mathbf{58.39 \pm 2.76}$ | $46.14 \pm 0.90$ |
| walker2d | 2 | $53.04 \pm 2.69$ | $\mathbf{79.83 \pm 2.29}$ | $58.50 \pm 4.25$ |
|  | 4 | $63.15 \pm 6.28$ | $\mathbf{83.42 \pm 5.43}$ | $77.41 \pm 2.37$ |
| halfcheetah | 2 | $74.08 \pm 2.33$ | $\mathbf{86.44 \pm 1.09}$ | $76.32 \pm 1.30$ |
|  | 4 | $68.35 \pm 2.60$ | $\mathbf{79.05 \pm 4.28}$ | $75.93 \pm 2.59$ |
| ant | 2 | $\mathbf{56.25 \pm 3.45}$ | $\mathbf{59.57 \pm 3.03}$ | $\mathbf{58.32 \pm 1.23}$ |
|  | 4 | $\mathbf{64.39 \pm 1.77}$ | $\mathbf{64.64 \pm 2.53}$ | $\mathbf{65.00 \pm 2.46}$ |

Table 9: Performance comparison of BC, our method (PALR) and direct action regularization version of PALR (PALR-ACT). The normalized scores averaged over the final 50 evaluations during training and we report mean and standard error over 5 different seeds. The method with the highest mean score and its competitive methods (within standard error) are highlighted in bold in each setting.

## D.2  Experiment on `carla-town`

D4RL benchmark provides two CARLA tasks: `carla-lane`, `carla-town`. As shown in Table 10, all algorithms fail to show meaningful imitation results, as none of them exceed random performance levels. This outcome is likely due to the dataset containing numerous instances of the same observations paired with different actions (59,984 observation-action pairs out of 100K pairs in total). Consequently, we have omitted the experiment results for `carla-town` from Table 1.

| Task | w | BC | KF | PrimeNet | RAP | FCA | MINE | PALR (Ours) |
|------|---|----|----|----------|-----|-----|------|-------------|
| carla-lane | 3 | $52.5 \pm 6.2$ | $66.6 \pm 2.1$ | $58.2 \pm 2.2$ | $25.3 \pm 5.4$ | $57.1 \pm 3.1$ | $60.1 \pm 4.1$ | $\mathbf{72.9 \pm 2.6}$ |
| carla-town | 3 | $-3.1 \pm 1.1$ | $-6.9 \pm 0.3$ | $-8.8 \pm 0.5$ | $-6.1 \pm 1.0$ | $-7.0 \pm 2.1$ | $\mathbf{1.0 \pm 0.8}$ | $-1.5 \pm 0.8$ |

Table 10: Performance evaluation of baseline and regularization methods on CARLA datasets provided by D4RL benchmark.

### D.3 Comparison with VIB (Variational Information Bottleneck)

To highlight the leakage of past action information is not trivially prevented by general representation regularization methods, we compare our approach with VIB [1], a representative method. We train BC with VIB with two different $\beta$ values, specifically, $\beta \in \{10^{-3}, 10^{-5}\}$, 5 seeds for each problem setting. whereas our method demonstrates significant improvements in 6 out of 8 problem settings.

| Task | w | BC | PALR (Ours) | VIB ($\beta = 10^{-3}$) | VIB ($\beta = 10^{-5}$) |
|---|---|---|---|---|---|
| hopper | 2 | $32.47 \pm 2.85$ | $\mathbf{42.01 \pm 2.44}$ | $30.57 \pm 1.61$ | $27.89 \pm 1.55$ |
| | 4 | $47.65 \pm 3.43$ | $\mathbf{58.39 \pm 2.76}$ | $51.20 \pm 2.18$ | $46.59 \pm 2.26$ |
| walker2d | 2 | $53.04 \pm 2.69$ | $\mathbf{79.83 \pm 2.29}$ | $44.28 \pm 2.32$ | $35.98 \pm 4.69$ |
| | 4 | $63.15 \pm 6.28$ | $\mathbf{83.42 \pm 5.43}$ | $67.78 \pm 2.67$ | $65.99 \pm 1.97$ |
| halfcheetah | 2 | $74.08 \pm 2.33$ | $\mathbf{86.44 \pm 1.09}$ | $73.76 \pm 2.401$ | $75.43 \pm 1.77$ |
| | 4 | $68.35 \pm 2.60$ | $\mathbf{79.05 \pm 4.28}$ | $67.89 \pm 1.52$ | $\mathbf{73.67 \pm 2.86}$ |
| ant | 2 | $\mathbf{56.25 \pm 3.45}$ | $\mathbf{59.57 \pm 3.03}$ | $\mathbf{59.54 \pm 1.65}$ | $55.29 \pm 4.27$ |
| | 4 | $\mathbf{64.39 \pm 1.77}$ | $\mathbf{64.64 \pm 2.53}$ | $\mathbf{61.63 \pm 2.77}$ | $\mathbf{61.25 \pm 2.88}$ |

Table 11: Performance comparison of BC, our method (PALR) and BC with VIB (Variational Information Bottleneck). The normalized scores averaged over the final 50 evaluations during training and we report mean and standard error over 5 different seeds. The method with the highest mean score and its competitive methods (within standard error) are highlighted in bold in each setting.

### D.4 Effectiveness on DT (Decision Transformer) policy

To assess the efficacy of our method in regularizing policies with complex network architectures, we conducted additional experiments employing the Decision Transformer (DT) [5], one of the prominent offline RL methods. Given that reward information is absent in the offline IL dataset, we inserted a reward input of 0 into DT's structure to retain its original configuration. Our regularization approach, termed DT-PALR, was applied to the last hidden state of DT. We evaluated both the standard DT and DT-PALR across 3 POMDP versions of MuJoCo tasks, consistent with the scenarios outlined in Table 1.

The results are presented in Table 12. With the exception of the `halfcheetah` task, it becomes evident that DT's performance lags behind that of BC, as demonstrated in Table 1. This divergence might be attributed to DT's utilization of a larger network size, rendering it more susceptible to capturing spurious causal relationships in tasks where access to complete states and rewards is restricted. Encouragingly, our results indicate a substantial enhancement in the performance of DT for the `hopper` and `walker2d` tasks when subjected to the DT-PALR method. This observation strongly suggests the adaptability of our approach to intricate architectures.

| Method | hopper | walker2d | halfcheetah |
|---|---|---|---|
| DT | $20.68 \pm 5.25$ | $23.93 \pm 4.13$ | $94.10 \pm 3.54$ |
| DT-PALR | $26.09 \pm 6.59$ | $32.58 \pm 5.63$ | $96.05 \pm 3.60$ |

Table 12: Effectiveness of PALR on Decision Transformer (DT) architecture. We train DT over 3 MuJoCo tasks based on our observation configurations and apply PALR to the last hidden state.

## D.5 Quantitative analysis on past action information leakage

We comprehensively evaluate HSCIC scores across 8 MuJoCo problem settings in Table 13. The results demonstrate that our method consistently reduces the conditional dependence. Notably, our method achieves lowest HSCIC scores in entire problem settings. Moreover, to show a more robust assessment of conditional dependence, we also estimate conditional mutual information (CMI). (see Table 14) As we described in Section 4.2.1, CMI can be decomposed into two MI terms, $I(a_{t-1}^E; \varphi_t, a_t^E)$ and $I(a_{t-1}^E; a_t^E)$, we estimate them using MINE [3] respectively. The table shows that our approach consistently presents lower CMI estimates compared to BC across 6 out of 8 problem settings.

| Task | w | BC | RAP | FCA | MINE | PALR (Ours) |
|------|---|-----|-----|-----|------|-------------|
| hopper | 2 | $1.783 \times 10^{-2}$ | $1.779 \times 10^{-2}(\downarrow)$ | $1.805 \times 10^{-2}(\uparrow)$ | $1.802 \times 10^{-2}(\uparrow)$ | $\mathbf{1.742} \times 10^{-2}(\downarrow)$ |
|  | 4 | $1.634 \times 10^{-2}$ | $1.617 \times 10^{-2}(\downarrow)$ | $1.629 \times 10^{-2}(\downarrow)$ | $1.650 \times 10^{-2}(\uparrow)$ | $\mathbf{1.587} \times 10^{-2}(\downarrow)$ |
| walker2d | 2 | $5.442 \times 10^{-2}$ | $5.523 \times 10^{-2}(\uparrow)$ | $5.415 \times 10^{-2}(\downarrow)$ | $5.482 \times 10^{-2}(\uparrow)$ | $\mathbf{5.362} \times 10^{-2}(\downarrow)$ |
|  | 4 | $5.279 \times 10^{-2}$ | $5.350 \times 10^{-2}(\uparrow)$ | $5.248 \times 10^{-2}(\downarrow)$ | $5.298 \times 10^{-2}(\uparrow)$ | $\mathbf{5.182} \times 10^{-2}(\downarrow)$ |
| halfcheetah | 2 | $4.010 \times 10^{-2}$ | $4.044 \times 10^{-2}(\uparrow)$ | $4.112 \times 10^{-2}(\uparrow)$ | $4.082 \times 10^{-2}(\uparrow)$ | $\mathbf{3.984} \times 10^{-2}(\downarrow)$ |
|  | 4 | $4.019 \times 10^{-2}$ | $4.052 \times 10^{-2}(\uparrow)$ | $4.092 \times 10^{-2}(\uparrow)$ | $4.069 \times 10^{-2}(\uparrow)$ | $\mathbf{3.968} \times 10^{-2}(\downarrow)$ |
| ant | 2 | $6.014 \times 10^{-2}$ | $6.164 \times 10^{-2}(\uparrow)$ | $6.182 \times 10^{-2}(\uparrow)$ | $6.178 \times 10^{-2}(\uparrow)$ | $\mathbf{6.008} \times 10^{-2}(\downarrow)$ |
|  | 4 | $5.938 \times 10^{-2}$ | $6.187 \times 10^{-2}(\uparrow)$ | $6.093 \times 10^{-2}(\uparrow)$ | $6.103 \times 10^{-2}(\uparrow)$ | $\mathbf{5.932} \times 10^{-2}(\downarrow)$ |

Table 13: $\widehat{\mathrm{HSCIC}}(a_t^I, a_{t-1}^E | a_t^E)$ evaluation results of regularization methods and baselines. HSCIC scores are measured on held-out datasets and averaged over the final 50 evaluations during training and statistics are calculated with 5 seeds. $(\downarrow), (\uparrow)$ marks whether the averaged HSCIC score of each method is lower than HSCIC score of BC or not.

| Task | w | BC | RAP | FCA | MINE | PALR (Ours) |
|------|---|-----|-----|-----|------|-------------|
| hopper | 2 | 0.1825 | 0.1826 $(\uparrow)$ | 0.1949 $(\uparrow)$ | 0.1891 $(\uparrow)$ | 0.1821 $(\downarrow)$ |
|  | 4 | 0.1489 | 0.1381 $(\downarrow)$ | 0.1481 $(\downarrow)$ | 0.1558 $(\uparrow)$ | 0.1422 $(\downarrow)$ |
| walker2d | 2 | 0.2685 | 0.2655 $(\downarrow)$ | 0.2798 $(\uparrow)$ | 0.2745 $(\uparrow)$ | 0.2705 $(\uparrow)$ |
|  | 4 | 0.2491 | 0.2353 $(\downarrow)$ | 0.2522 $(\uparrow)$ | 0.2461 $(\downarrow)$ | 0.2411 $(\downarrow)$ |
| halfcheetah | 2 | 0.1562 | 0.1496 $(\downarrow)$ | 0.1612 $(\uparrow)$ | 0.1519 $(\downarrow)$ | 0.1509 $(\downarrow)$ |
|  | 4 | 0.1396 | 0.1530 $(\uparrow)$ | 0.1551 $(\uparrow)$ | 0.1484 $(\uparrow)$ | 0.1502 $(\uparrow)$ |
| ant | 2 | 0.1579 | 0.1623 $(\uparrow)$ | 0.1743 $(\uparrow)$ | 0.1567 $(\downarrow)$ | 0.1374 $(\downarrow)$ |
|  | 4 | 0.1585 | 0.1653 $(\uparrow)$ | 0.1642 $(\uparrow)$ | 0.1708 $(\uparrow)$ | 0.1450 $(\downarrow)$ |

Table 14: $\widehat{I}(a_t^I; a_{t-1}^E | a_t^E)$ evaluation results of regularization methods and baselines. Since the conditional MI can be decomposed into two MI terms as described in Section 4.1.1, we estimate two terms with MINE estimator respectively. We use the policy of the last iteration for each method to generate policy actions. $(\downarrow), (\uparrow)$ indicates whether the averaged CMI score of each method is lower than the score of BC or not.

### D.6 Details on the leakage-performance correlation

We first clarify the relationship between the number of training data and HSCIC scores in Section 5.1. To clarify, we plot the correlation into Figure 4. The correlation indicates that when the training data is insufficient, the problem of past action information leakage becomes more severe. This phenomenon can be attributed to the higher risk of overfitting in cases with smaller training instances, which can lead to the capture of false causal relationships within the training data. These findings align with similar results reported in [8] (see Figure 4 in [8]).

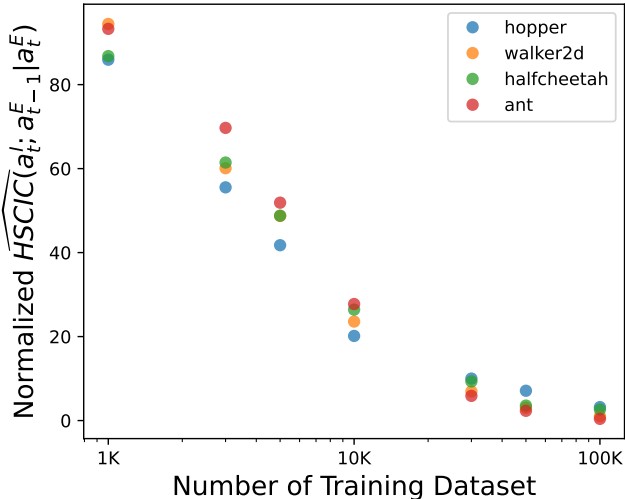

Figure 4: Relationship between the number of training data and the normalized HSCIC score.

Furthermore, to provide more insights for understanding Figure 1, we plot all points that indicate the normalized HSCIC and the normalized score of each BC policy in Figure 5. In addition, we calculate the sample Pearson's correlation coefficients using every point of each task (the first 4 columns) and all tasks (the last column) on Table 15.

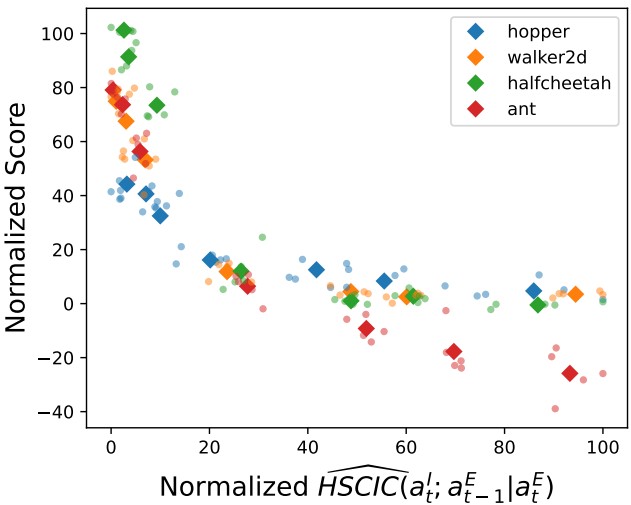

Figure 5: Scatterplot that shows a correlation between HSCIC estimates and performance. Each small point indicates HSCIC and the performance of each policy, and the diamond point (◇) indicates the mean point of each group (which consists of policies trained with the same dataset size).

| hopper | walker2d | halfcheetah | ant | Total |
|--------|----------|-------------|-----|-------|
| -0.8358 | -0.8163 | -0.8564 | -0.9273 | -0.8130 |

Table 15: Sample Pearson's correlation coefficients between the normalized HSCIC and the normalized score.

## E  Broader Impacts

This work primarily addresses the reduction of the pervasive past action leakage effect in imitation learning with observation histories. Notably, our findings suggest potential vulnerabilities that may be exploited by adversarial actors. For instance, malicious attackers could introduce the conditional dependencies into the system, leading to potentially catastrophic outcomes, such as repeating the same actions. This will produce potentially irreversible consequences on the system, especially for security-critic tasks such as autonomous driving and medical devices, or industrial automation. Therefore, understanding and mitigating this risk is crucial for the safe and efficient operation of such systems.

## F  Limitations

While our approach is easily applicable to real-world tasks, the range of experiments in this paper is limited to robotics simulation. A broader scope of experiments, including more realistic observation settings like image-based or noise-inclusive observations, could provide a stronger argument for the effectiveness and real-world applicability of our method. Additionally, our method assumes that control-relevant information is sufficiently captured in the observation histories. However, this assumption might not hold in some real-world scenarios where key control information is missing and not retrievable from the observation histories. In such cases, the learning model requires additional information, inductive bias or expert knowledge to function effectively.

## G  Computation Resources

For our experiments, we used a cluster system with 20 nodes that have the following system specs:

- CPU: Intel i7-9700K CPU (3.60GHz)

- GPU: TITAN Xp (VRAM 12 GB)

## H  Licenses

Our code has been developed based on publicly available code repositories released under MIT licenses. The code used to generate our experimental results also follows to the MIT license. For more detailed information, see the `README.md` and `LICENSE` files included in our code files.

