# OpenReview forum: "Regularized Behavior Cloning for Blocking the Leakage of Past Action Information"
_NeurIPS.cc/2023/Conference — NeurIPS 2023 spotlight_

### Official Review · Reviewer_Xa8t · 2023-06-29

**Soundness:** 2 fair
**Presentation:** 3 good
**Contribution:** 2 fair
**Rating:** 5
**Confidence:** 3

**Summary:**

This paper introduces Past Action Leakage Regularization (PALR) for blocking the leakage of past action information during behavior cloning. Concretely, PALR focuses on the problem when a BC agent simply remembers the past actions rather than learning a generalized behavior, which gives a degenerated policy. Specifically, PALR tackles the problem by learning a representation of the history to remove unnecessary from the observations via conditional independence. PALR adopts the HSCIC metric to measure the conditional independence, which alleviates the shortcomings of the information-theoretic metric. In their experiments, the authors compared different metrics for behavior cloning on a range of continuous control tasks.

**Strengths:**

This paper is overall well-written and easy to follow. The idea is well-motivated as well.

**Weaknesses:**

The paper has the following main issues.

Firstly, the contribution of this paper is relatively incremental. The formulation of the BC regularization as conditional entropy is new. Wen et al. [1] was the first to propose this formulation. As a result, the main contribution of this work is only the HSCIC regularization term.

In addition, the paper has claimed that the focus of the work is POMDPs, where the past action leakage might come from the history. However, in the experiments, the only domain considered is state-based continuous control, which is a fully observable MDP. In my opinion, such a domain does not support the claim of the work, and is unsuitable to measure the capability of the proposed method. More complex and natural POMDP domains should be considered, e.g., indoor navigation with pixel-based observations, to better support the claims.

References:

[1] Wen, C., Lin, J., Darrell, T., Jayaraman, D., & Gao, Y. (2020). Fighting copycat agents in behavioral cloning from observation histories. Advances in Neural Information Processing Systems, 33, 2564-2575.


**Questions:**

How does the method compare with more advanced imitation learning algorithms that require interactions with the environment, e.g., DAC [1] and PWIL [2]? BC is a relatively weak baseline. Can this regularization term be used in these algorithms to further boost the performance?

References:

[1] Kostrikov, I., Agrawal, K. K., Dwibedi, D., Levine, S., & Tompson, J. (2018). Discriminator-actor-critic: Addressing sample inefficiency and reward bias in adversarial imitation learning. arXiv preprint arXiv:1809.02925.

[2] Dadashi, R., Hussenot, L., Geist, M., & Pietquin, O. (2020). Primal wasserstein imitation learning. arXiv preprint arXiv:2006.04678.

**Limitations:**

This paper has relatively limited novelty and insufficient experiment results. Given its current state, I think the paper is not ready for publication yet.

---

> ### Author Rebuttal · Authors · 2023-08-10
>
> We deeply appreciate your constructive and insightful comments.
>
> **1. Novelty in our work**
>
> We think that although the HSCIC regularization indeed holds significance within our work, our contribution extends beyond the introduction of the HSCIC regularization term: in the paper, we introduced a novel perspective on the issue of past action leakage through the lens of conditional independence. Our approach, which is derived naturally from this perspective and is implemented as a regularizer for conditional independence, covers FCA [A], HSCIC, and all the other measures for conditional independence. Note that an analogous approach based on conditional entropy, not conditional independence, is less general; for instance, it does not cover HSCIC. Identifying the important role of conditional independence in our problem setting is, we believe, an important conceptual contribution of our work.
>
>
> **2. Mujoco experiments in POMDP settings**
>
> We want to clarify that we have reconfigured the observations of Mujoco tasks to retain only positional information while excluding velocity information from observations, hence we conduct experiments on POMDP versions of Mujoco tasks. For further details, please refer to Section C.1 of the supplementary material.
>
>
> **3. Additional experiment on a complex domain**
>
>  To see if our approach is effective on a complex domain, we perform an evaluation of our approach and the baseline methods within the CARLA environment. Please see our general response and Table A in the attached PDF file for more details.
>
> **4. BC as a fundamental baseline**
>
> Our focus is on offline IL-OH, excluding online interaction with the environment. While BC is simple, it's not weak in this context. In our experiments, BC's performance rivaled that of other baselines like FCA and MINE. Additionally, BC's role in simplifying training, away from intricate policy optimization, helps dissect the impact of different regularization methods.
>
> **5. Comparison with online imitation learning algorithms**
>
>  We appreciate your comments and would like to provide clarification regarding our research focus. Our research centers on offline IL from observation histories (IL-OH), and while we acknowledge methods like DAC and PWIL focus on online IL with fully-observable MDPs, which is beyond the scope of our present research. Please also note that our study pertains to addressing the challenge from the problematic phenomenon of past action information leakage, which is present in offline IL-OH problems. If such a phenomenon occurs in online IL, we believe that our regularization could effectively mitigate the spurious correlation between past expert actions and imitator actions. We leave this extension as our future research.
>
>
> **References**
>
> [A] Wen et al., "Fighting Copycat Agents in Behavioral Cloning from Observation Histories.", NeurIPS 2020.

---

> > ### Comment · Reviewer_Xa8t · 2023-08-15
> >
> > I appreciate the author's rebuttal and efforts for additional experiments. They have addressed most of my concerns. I've changed my rating to 5 and as this paper possesses the necessary qualities for acceptance.

---

> > > ### Author Response · Authors · 2023-08-15
> > >
> > > Thank you for your positive response. We are glad to hear that your concerns are addressed.

---

### Official Review · Reviewer_xrec · 2023-07-06

**Soundness:** 3 good
**Presentation:** 3 good
**Contribution:** 3 good
**Rating:** 5
**Confidence:** 2

**Summary:**

This paper solves the past action information leakage problem in behavior cloning. The paper first formally defines this problem, and then provides some potential regularization methods. After careful analysis of the pros and cons of each method, the authors decide to use HSCIC and validate its performance on several tasks.

**Strengths:**

- The flow of this paper is very clear: problem definition -- potential solutions -- analysis -- evaluation
- In the evaluation, the authors not only show the improvement of the final performance of BC but also highlight the correlation between the performance and HSCIC regularization. This strongly proves that regularization is the major contribution to performance improvement.
- Comprehensive ablation study and analysis are provided in the paper.

**Weaknesses:**

The major concern is how to apply this regularization to complex environments with images (or even multi-modality) as input and high-dimensional action space (6 DoF or even more). If the proposed method can be applied to such complex tasks, I'm willing to increase my score.

**Questions:**

The BC policy used in this paper is just a feed-forward policy p(a|s).
Is it possible to apply the regulation to other kinds of policy, for example, an energy-based model [1] or even a diffusion model[2]?

[1] Florence, Pete, et al. "Implicit behavioral cloning." Conference on Robot Learning
[2] Chi, Cheng, et al. "Diffusion policy: Visuomotor policy learning via action diffusion." arXiv preprint arXiv:2303.04137

**Limitations:**

no explicit limitation discussion are provided in the paper

---

> ### Author Rebuttal · Authors · 2023-08-10
>
> We hold your thoughtful viewpoints on our work in high regard.
>
> **1. Effectiveness in complex task**
>
> To validate past action leakage regularization is effective on complex tasks, we conduct an experiment on CARLA environment. Please see our general response and Table A of PDF.
>
> **2. Applicability to complex policy architecture**
>
> We appreciate your consideration of different policy models. While we chose a feed-forward policy due to task generality, we acknowledge the potential to extend our approach to more specialized policies. Methods like energy-based [1] and diffusion models [2] designed for tasks with visual inputs and complex action spaces could enhance the applicability of our approach. Transitioning to these models is a direction we foresee exploring in adapting PALR for visuomotor control tasks. Furthermore, we've extended our investigations to the Decision Transformer (DT) [A] architecture to assess PALR's effectiveness on complex policies, as detailed in our general response and Table B of PDF.
>
> **3. Limitation Section**
>
> Limitations section can be found in Section F provided in supplementary material.
>
> **Reference**
>
> [A] Chen et al., “Decision Transformer: Reinforcement learning via Sequence Modeling.”, NeurIPS 2021.

---

### Official Review · Reviewer_UXMA · 2023-07-07

**Soundness:** 3 good
**Presentation:** 4 excellent
**Contribution:** 3 good
**Rating:** 7
**Confidence:** 5

**Summary:**

This paper proposes to use HSCIC (Hilbert-Schmidt Conditional Independence Criterion) to alleviate the leakage of information from past actions in behavior cloning from observation histories. The advantage of HSCIC compared with information-theoretic regularization is it can compute in closed-form and does not require parametric assumption on the data distribution.

Experimental results show HSCIC is a good indicator of agent performance and by directly minimizing HSCIC, it consistently improves over naive BC and other baselines with different regularizations.

**Strengths:**

Experimental results show significant improvement over baselines across different locomotion tasks.
Good analysis showing negative correlation between HSCIC and agent performance, which motivates to minimize it directly in training objectives.
Writing is clear and straightforward. Good comparison of kernel-based regularization and information-theoretic approaches.

**Weaknesses:**

Lack of comparisons with non-regularization based approaches to tackle past action information leakage. For example, Key-frame focused visual imitation learning proposes a sampling/weighting approach and Fighting fire with fire: Avoiding DNN shortcuts through priming proposes to use additional prior information as inductive biases.
Experiments are all on simple state-based results. Hard to judge the performance on more complex domains, e.g. visual imitation learning, robotic manipulation or autonomous driving.
Kernel-based approaches have time complexity depending on the number of samples. It would be worth comparing the computation tradeoff between different approaches.

**Questions:**

How well does the proposed methods work on more complex environments, e.g. Atari, Carla, etc.?
Could you show some quantitative results on how the agents avoid failure due to leakage of previous action?
Is there any heuristics of turning the hyperparameter alpha?
Could this regularization be combined with other approaches (e.g. information bottleneck, sampling) to give better results?

**Limitations:**

Authors state theirhyperparameters are sensitive and hard to tune without interaction with the environment. No broader impact section found in the main paper.

---

> ### Author Rebuttal · Authors · 2023-08-10
>
> We value your considerate insights and invaluable suggestions concerning our work.
>
> **1. Comparisons with other non-regularization approaches**
>
> Thank you for the pointers to the missing important related work. We recognize that both of your suggestions are valid to compare with our approach. During the remaining rebuttal period, we do our best to evaluate these two approaches and update our experiment.
>
> **2. Experiments in complex domain**
>
> We additionally conduct evaluation of our method and baselines in CARLA environment provided in D4RL dataset. Please see our general response and Table A of PDF file.
>
> **3. Time complexity comparison**
>
> It is worth noting that the computation time of our PALR implementation increases with mini-batch sizes (denote $M$), but it does not depend on the number of total data samples; the kernel regression is applied to each mini-batch, not to the entire training set. Furthermore, there are techniques for computing the HSCIC estimator efficiently [A].
>
>  In contrast, current information-theoretic approaches often require multiple iterations of inner optimization (denote $k$ as the number of inner iterations) to obtain reliable estimations of quantities of interest (e.g. mutual information or entropy). In practice, the elapsed time of these approaches can be significantly higher than PALR. To compare computation costs, we measure the wall-clock time of one policy update over our method and baselines. ($M$= 1024, $k$=5, 1000 repetitions)
>
> |  | BC | RAP | FCA | MINE | PALR |
> | --- | --- | --- | --- | --- | --- |
> | Big-O | $O(M)$ | $O(M)$ | $O(kM)$ | $O(kM)$ | $O(M^3)$ |
> | Average Elapsed time (ms) | 3.038 | 5.295 | 12.203 | 19.854 | 15.64 |
>
> [Machine specification] CPU : Intel(R) Core(TM) i7-4770 @ 3.40GHz (4 cores), GPU : Titan X,  Memory : 32Gb
>
>  Our approach demonstrates scalable time complexity, particularly with moderate batch sizes, for example 512 to 1024. This ensures that PALR remains scalable and effective in various practical scenarios. We will include a detailed discussion of the computation tradeoff comparison in the appendix of our revised manuscript.
>
>
> **4. Quantitative analysis**
>
> To enhance the credibility of our experimental interpretation, we broaden our quantitative analysis. Firstly, we assess the HSCIC score across all 8 problem settings (refer to Table C in our provided PDF). Secondly, we examine conditional MI as an additional measure of conditional independence to robustly validate our argument (refer to Table D in the PDF). Please see our general response for further details. If you have any further question in additional quantitative results, please feel free to inform us.
>
>
> **5. Heuristics of tuning alpha**
>
> In the context of offline model selection for sequential decision-making, it is well-acknowledged that identifying optimal hyperparameters from offline dataset is a non-trivial task [B,C,D,E]. Systematic hyperparameter tuning in an offline setting remains an open challenge. In our empirical observations from Mujoco experiments, we note that effective coefficients typically fall within the range of 10 to 1000. Notably, this range tends to align the scale of the HSCIC regularization term with that of the BC loss term.
>
> **6. Combination with other approaches**
>
> PALR could potentially synergize with orthogonal regularization approaches, which we consider for future exploration.
>
>
> **References**
>
> [A] Quinzan et al., “Learning Counterfactually Invariant Predictors”, arxiv 2022.
>
> [B] Hussenot et al., “Hyperparameter Selection for Imitation Learning.”, ICML 2021.
>
> [C] Zhang et al. "Towards Hyperparameter-free Policy Selection for Offline Reinforcement Learning." NeurIPS 2021.
>
> [D] Paine et al. "Hyperparameter Selection for Offline Reinforcement Learning." arXiv 2020.
>
> [E] Lee et al., “Batch Reinforcement Learning with Hyperparameter Gradients.”, ICML 2020.

---

> > ### Author Response · Authors · 2023-08-14
> > **Update in Comparisons with Non-regularization-based Approaches**
> >
> > To address your concern regarding the lack of comparisons with non-regularization-based approaches, we have introduced additional baseline methods. As you suggested, we have included two additional baselines [F, G] and expanded our evaluation to cover all tasks included in our manuscript and rebuttal (4 POMDP versions of Mujoco tasks and the CARLA environment). The results are summarized in the following table:
> >
> > | Task | W | BC | RAP | FCA | MINE | PALR | Keyframe [F] | PrimeNet [G] |
> > | --- | --- | --- | --- | --- | --- | --- | --- | --- |
> > | hopper | 2 | 32.47 ± 2.85 | 20.19 ± 1.38 | 31.89 ± 2.54 | 24.98 ± 1.89 | **42.01 ± 2.44** | 32.01 ± 1.86 | 29.98 ± 1.57 |
> > |  | 4 | 47.65 ± 3.43 | 32.61 ± 2.62 | 36.90 ± 2.35 | 37.60 ± 3.14 | **58.39 ± 2.76** | 45.74 ± 0.95 | 45.31 ± 2.77 |
> > | walker | 2 | 53.04 ± 2.69 | 15.82 ± 2.03 | 63.11 ± 2.69 | 58.62 ± 5.52 | **79.83 ± 2.29** | 49.97 ± 2.32 | 48.50 ± 3.25 |
> > |  | 4 | 63.15 ± 6.28 | 25.39 ± 2.14 | **81.88 ± 3.26** | 68.71 ± 6.66 | **83.42 ± 5.43** | 77.37 ± 1.97 | **79.17 ± 3.30** |
> > | halfcheetah | 2 | 74.08 ± 2.33 | 63.90 ± 2.14 | 78.24 ± 2.80 | 76.29 ± 1.87 | **86.44 ± 1.09** | 64.26 ± 1.41 | 61.47 ± 1.90 |
> > |  | 4 | 68.35 ± 2.60 | 58.97 ± 2.66 | 69.89 ± 2.64 | 73.4 ± 2.35 | **79.05 ± 4.28** | 55.71 ± 4.14 | 45.51 ± 1.66 |
> > | ant | 2 | 56.25 ± 3.45 | 44.05 ± 1.19 | 51.08 ± 2.19 | 53.88 ± 1.87 | **59.57 ± 3.03** | 54.94 ± 1.68 | 51.72 ± 2.38 |
> > |  | 4 | **64.39 ± 1.77** | 48.63 ± 2.63 | 57.73 ± 1.25 | 56.56 ± 1.76 | **64.64 ± 2.53** | 48.59 ± 3.75 | 58.18 ± 1.92 |
> > | carla-lane | 3 | 53.82 ± 7.66 | 20.15 ± 7.91 | 51.83 ± 7.91 | 60.62 ± 6.40 | **72.27 ± 2.62** | 66.36 ± 3.44 | 61.72 ± 1.53 |
> > | carla-town | 3 | -3.94 ± 1.68 | -7.64 ± 0.97 | -9.42 ± 0.47 | **0.13 ± 0.93** | -1.15 ± 1.01 | -6.68 ± 0.07 | -9.50 ± 0.33 |
> >
> > For implementation details of [F], we used a softmax function as a weighting function for the weighted BC in Mujoco tasks and a step function in CARLA tasks, akin to the original paper’s experiment. Similar to regularization-based methods, we selected optimal hyperparameters $\tau \in [0.01, 0.1, 1, 10, 100]$, which represents the temperature for the softmax function. For CARLA tasks, we used the same hyperparameters (threshold=0.1,weight=5) as in the CARLA experiment of the original paper. Please refer to the reply to our general response for other details.
> >
> > **References**
> >
> > [F] Wen et al., “Keyframe-Focused Visual Imitation Learning”, ICML 2021.
> >
> > [G] Wen et al., “Fighting Fire with Fire: Avoiding DNN Shortcuts through Priming”, ICML 2022.

---

> > > ### Comment · Reviewer_UXMA · 2023-08-19
> > >
> > > Thank you for the great rebuttal! Most of my concerns are addressed!
> > > I raise my score to 7.

---

> > > > ### Author Response · Authors · 2023-08-20
> > > >
> > > > We’re happy to hear that your concerns are addressed. Thank you for your positive comments!

---

### Official Review · Reviewer_TkkQ · 2023-07-13

**Soundness:** 3 good
**Presentation:** 4 excellent
**Contribution:** 3 good
**Rating:** 8
**Confidence:** 3

**Summary:**

This paper aims to solve the problem of negative action leakage from past observations in the context of imitation learning. This is specifcally applicable to imitation learning when considering a history of observations. The paper argues that an information theoretic (entropy or MI based) regularization requires training an additional network and a nested optimization, however, HSCIC avoids both of these inefficiencies. The experiments show that (1) HSCIC is a good indicator of leakage, and (2) the propose approach which regularizes the BC loss with HSCIC outperforms other information theoritic and vanilla baselines quite a bit, in offline imitation settings.

**Strengths:**

- To my knowledge this is a novel approach
- The method is very interesting, and explained in a way that is easy to understand, and is also an innovative solution
- The motivation is well described, and this is an important area to explore
- The results show strong performance, and the analysis of the utility of HSCIC is insightful as well
- The paper is well written

In general, I believe this paper should be presented at the conference.

**Weaknesses:**

I think a couple of things could have been discussed:

- How sequence modelling architectures can play a role here such as transformer/Decision Transformer (Chen et al., 2021)
- What are the different modes of leakage, and do they affect performance in different ways


**Questions:**

See weaknesses

**Limitations:**

Limitations are sufficienlty addressed

---

> ### Author Rebuttal · Authors · 2023-08-10
>
> We appreciate your positive feedback.
>
> **1. Applicability to sequence modeling architecture**
>
> We acknowledge the potential effectiveness of our approach in sequence modeling architecture. To explore this further, we have applied our regularization to the Decision Transformer architecture you suggested. Please refer to our general response and Table B in the provided PDF.
>
> **2. Different modes of leakage on imitation learning**
>
> The exploration of information leakage in imitation learning remains a promising direction of research, accompanied by several open questions. One example can be drawn to the concept of target leakage [A] in the context of data mining, where information produced from targets is contained in input data. Similarly, in imitation learning, a related phenomenon arises where information generated by the current expert action can become embedded within the observation. In this situation, the imitator easily captures that spurious information to predict expert action during training, but the imitator will fail in test time. This discrepancy between training and inference will lead to harmful effects on performance.
>
> **Reference**
>
> [A] Kaufman et al., "Leakage in Data Mining: Formulation, Detection, and Avoidance.", KDD 2011.

---

### Official Review · Reviewer_jVBb · 2023-07-22

**Soundness:** 3 good
**Presentation:** 3 good
**Contribution:** 3 good
**Rating:** 5
**Confidence:** 3

**Summary:**

This paper addresses the information leakage problem of imitation learning with observation histories. To this end, the paper measures the leakage of past action information based on conditional independence and proposes Past Action Leakage Regularization (PALR) for behavioral cloning (BC). The experiments show that the proposed method outperforms four baselines on four MuJoCo continuous control environments. Ablation studies suggest PALR can improve BC with a proper coefficient. This work defines and studies an essential problem of imitation learning from partially observable environments. However, the experiments can be improved to make the work more concrete.

**Strengths:**

**Clarity**
- The overall writing is clear. The paper gives clear descriptions in both theoretical and intuitive ways. The notations, formulations, and theorems are well-explained.

**Ablation study**
- Ablation studies are comprehensive. The provided ablation studies help understand the effectiveness of the regularization coefficient (Sec.5.2) and the target to apply regularization (Sec.D).

**Reproducibility**
- The code is provided, which helps understand the details of the proposed framework.
- Given the clear description in the main paper and the details provided in the supplementary materials, I believe reproducing the results is possible.

**Weaknesses:**

**Method**
- This work proposes a kernel-based method to regularize BC for imitation learning from partially observable environments. While I am aware of the advantages of kernel-based methods (stable, does not need additional networks or hyperparameters tuning), recent works use neural networks, such as variational model [1] or causal transformer [2], to capture essential information from observation histories and have shown promising results.
The effectiveness of the proposed PALR would be more convincing if the authors could demonstrate comparisons to the above methods.

**Experiments**
- Some parts of the experimental results are not easily interpretable. Figure 1 shows the negative correlation between the number of training data and the HSCIC score. Can the author explain why the information leakage problem is more severe when the training data is insufficient?

- I am not entirely convinced by the explanation of why FCA [3] and MINE [4] get inferior results (Line 340-345 & Figure 2a).
The author only evaluates the HSCIC score of each method on hopper-W4. However, the proposed PALR is directly regularized by the HSCIC metric, so it is straightforward that the proposed PALR gets the lowest HSCIC score.
It would be better to evaluate the HSCIC score on all environment setups in Table 1 to 1) support the negative correlation between HSCIC estimations and the performance of algorithms and 2) analyze the performance of FCA and MINE, which outperform BC only on walker2d and halfcheetah environments.
- The paper only evaluates the proposed method on four continuous control environments, which is insufficient. RL tasks such as navigation, robot arm manipulation, or Atari games can also be considered.

**Experiment details**
- The details of the normalized score (line 280) are missing. What is the maximum score and the minimum score?

**Typo**
Line 323: ..., which is a one of the ...

[1] Rafailov, R., Yu, T., Rajeswaran, A., & Finn, C. (2021). Visual adversarial imitation learning using variational models. Advances in Neural Information Processing Systems, 34, 3016-3028.
[2] Bonatti, R., Vemprala, S., Ma, S., Frujeri, F., Chen, S., & Kapoor, A. (2022). Pact: Perception-action causal transformer for autoregressive robotics pre-training. arXiv preprint arXiv:2209.11133.
[3] Wen, C., Lin, J., Darrell, T., Jayaraman, D., & Gao, Y. (2020). Fighting copycat agents in behavioral cloning from observation histories. Advances in Neural Information Processing Systems, 33, 2564-2575.
[4] Belghazi, M. I., Baratin, A., Rajeshwar, S., Ozair, S., Bengio, Y., Courville, A., & Hjelm, D. (2018, July). Mutual information neural estimation. In International conference on machine learning (pp. 531-540). PMLR.

**Questions:**

See above

**Limitations:**

See above

---

> ### Author Rebuttal · Authors · 2023-08-10
>
> We are grateful for your constructive and enlightening comments.
>
> **1. Comparison to other methods**
>
> We appreciate your comments on alternative methods and their potential effectiveness. However, we would like to clarify some distinctions between the alternative methods in [1] and [2] and our proposed PALR.
> Firstly, the variational model [1] is designed for online IL scenarios, where the agent can interact with the environment during the learning process. In contrast, we focus on offline IL, where the policy is trained solely on a pre-collected dataset without any interaction with the environment. Consequently, the direct application of the variational model to our offline setting is not straightforward.
> Similarly, the causal transformer [2] involves a domain-specific fine-tuning step in each robotics downstream task, making it challenging to perform a fair comparison with our PALR, which is designed to be domain-independent. We specifically designed PALR to be applicable to a broader range of offline IL from observation history, with a focus on addressing past action leakage.
> While we recognize the potential effectiveness of the above two methods, we highlight that PALR is a generic approach for offline IL that does not rely on any domain-specific information. Our regularization method aims to improve imitation performance in scenarios where past action leakage remains a challenge, making it applicable across various domains without requiring domain-specific considerations.
>
> **2. Correlation between dataset size and HSCIC**
>
> Thank you for pointing out the negative correlation between the number of dataset and HSCIC score in Figure 1 in our manuscript. The correlation indicates that when the training data is insufficient, the problem of past action information leakage becomes more severe. To clarify, we plot the correlation into Figure A in the PDF file.
> This phenomenon can be attributed to the higher risk of overfitting in cases with smaller training instances, which can lead to the capture of false causal relationships within the training data. These findings align with similar results reported in [A] (see Figure 4 of the paper).
>
> **3. Complete evaluation on conditional independence**
>
> First, we clarify that $\mathrm{HSCIC}(\varphi_t, a^E_{t-1} | a^E_{t})$ and $\mathrm{HSCIC}(a^I_t, a^E_{t-1} | a^E_{t})$ (the score recorded in Figure 2a) are different. PALR regularizes HSCIC with respect to representation, not HSCIC with respect to imitator action directly. As shown in Theorem 1, PALR regularization enforces the conditional independence between actions.
> To provide a more trustful interpretation of our experiment, we comprehensively expand our quantitative analysis: (1) We evaluate the HSCIC score in all 8 problem settings (see Table C of our PDF file.) (2) We evaluate conditional MI as another conditional independence metric to provide robust verification of our argument. (see Table D of PDF) Please refer to our general response.
>
> **4. Evaluation on complex task**
>
> We perform an evaluation of our approach and the baseline methods in the CARLA environment using the D4RL dataset. Please see our general response and Table A of PDF.
>
> **5. Normalized score**
>
> We follow the D4RL dataset evaluation protocol, which designates the expert score as the upper limit and the score of a random agent as the lower limit for normalization. We will clarify this in the revision.
>
> **Reference**
>
> [A] Haan et al., “Causal Confusion in Imitation Learning.”, NeurIPS 2019.

---

> > ### Comment · Reviewer_jVBb · 2023-08-10
> > **Thanks for the rebuttal**
> >
> > I appreciate the author's rebuttal, which addresses some of my concerns. I believe this work studies a promising problem and provides meaningful insights. However, I still feel the experiments are a bit limited. As suggested by Reviewer xrec, comparing the proposed method against implicit BC and diffusion policies can make this work more convincing. Also, as described in my review, experimenting with robot arm/dexterity manipulation tasks or games can significantly widen the scope of experiments. In sum, I am still slightly leaning toward accepting this paper while I won't fight for this paper if the majority of the reviewers have a different opinion.

---

> > > ### Author Response · Authors · 2023-08-14
> > > **Addressing Concerns in Our Experiments**
> > >
> > > Thank you for your thoughtful feedback for our rebuttal.
> > >
> > > As discussed in our response to Reviewer xrec, we want to emphasize that our approach focuses on a general solution for offline IL-OH, not specifically designed to resolve challenges posed by high-dimensional action spaces, which are commonly addressed by methods like implicit BC [B] or diffusion policies [C]. Acknowledging the validity of your suggestion, a comparison with such methods could indeed demonstrate our method's adaptability to high-dimensional action spaces. However, due to limited computational resources, we have predominantly focused on evaluating our approach across standard tasks, leaving this research direction open for future exploration.
> > >
> > > Instead, to see if our method can be effectively applied to complex policy structures (e.g. diffusion models or transformers, …) with high capacity enough to cover various imitation tasks, we present experiment results of applying our regularization method to the Decision Transformer [D] as a representative instance of complex policies. Details can be found in paragraph 2 of our general response and Table B within the attached PDF.
> > >
> > > Furthermore, to extend the coverage of our experiment comparison, we have conducted additional evaluations on (1) pixel-based imitation tasks (CARLA experiment) and (2) comparison of two additional baselines [E,F] across all our tasks. Please see our comment to the general response.
> > >
> > > We hope these expanded results can address your remaining concerns regarding our work.
> > >
> > > **Reference**
> > >
> > > [B] Florence, Pete, et al. "Implicit Behavioral Cloning.", Conference on Robot Learning 2021.
> > >
> > > [C] Chi, Cheng, et al. "Diffusion Policy: Visuomotor Policy Learning via Action Diffusion.", arXiv 2023.
> > >
> > > [D] Chen et al., “Decision Transformer: Reinforcement learning via Sequence Modeling.”, NeurIPS 2021.
> > >
> > > [E] Wen et al., “Keyframe-Focused Visual Imitation Learning.”, ICML 2021.
> > >
> > > [F] Wen et al., “Fighting Fire with Fire: Avoiding DNN Shortcuts through Priming.”, ICML 2022.

---

### Official Review · Reviewer_4vtt · 2023-07-25

**Soundness:** 3 good
**Presentation:** 3 good
**Contribution:** 3 good
**Rating:** 7
**Confidence:** 5

**Summary:**

This paper proposes Past Action Leakage Regularization (PALR) to resolve the copycat problem in behavior cloning (BC) methods:

1. mathematically defines the past-information-leakage problem.

2. introduces PALR to formalize the methods.

3. uses Hilbert-Schmidt Conditional Independence Criterion(HSCIC) to measure the conditional independence, and analyses the advantages of HSCIC over CMI as the metric.

4. provides experiments verifying the correlation between HSCIC and performance, and the superiority of the HSCIC-based method over baseline methods.


**Strengths:**

1. The paper is well-written and easy to follow in most parts.

2. The authors provide sufficient experiments to verify their claims.

3. The idea of replacing CMI with HSCIC is natural in maths and easy to apply. It does bring a big improvement over the previous methods.

4. The paper comprehensively explains how HSCIC becomes a better metric than CMI.



**Weaknesses:**

1. Some small typos: $H_{\mathcal{X}}$ in line 133 should be $\mathcal{H}_{\mathcal{X}}$; the full name of MOMDP should use \emph{observable} other than \emph{observed} in line 153. I didn't go through every word, so there may be other mistakes.

2. In line 203, $\phi_t=\phi(z_{t_\omega:t})$, but in line 210, 212, $\phi\sim\phi(z_{t_\omega:t})$. This caused some confusion in reading. Did I miss anything?

3. In line 193, ''Ideally we would like to achieve conditional independence in Eq. (2)" is not obvious. And actually, it's not reasonable to require eq (2) to be true, for even a perfect imitator can't do this. The structure of MOMDP described in section 3.2 determines that it's not mathematically possible to predict $A_t^I$ given $A_t^E$. A better objective may be minimizing the CMI, as is practically done in the experiment. Furthermore, I know that this has been an unsolved problem from the first time copycat problem being proposed, but you may modify it by coming up with a better lower bound of CMI or HSCIC other than 0 to fix this mathematical flaw.

4. Theorem 1 does not help much in proving the claim that lower loss indicates lower dependence, which only proves the equality between two independences. I believe that this is true and thus PALR is effective. Can you come up with a new theorem to fix this?

5. Some important references are missing:

[1] Ortega P A, Kunesch M, Delétang G, et al. Shaking the foundations: delusions in sequence models for interaction and control[J]. arXiv preprint arXiv:2110.10819, 2021.

[2] Wen C, Lin J, Qian J, et al. Keyframe-Focused Visual Imitation Learning[C]//International Conference on Machine Learning. PMLR, 2021: 11123-11133.

[3] Wen C, Qian J, Lin J, et al. Fighting fire with fire: Avoiding dnn shortcuts through priming[C]//International Conference on Machine Learning. PMLR, 2022: 23723-23750.

[4] Spencer J, Choudhury S, Venkatraman A, et al. Feedback in imitation learning: The three regimes of covariate shift[J]. arXiv preprint arXiv:2102.02872, 2021.

**Questions:**

1. Do you have any explanation for the phenomenon that more historical information damages the performances in HalfCheetah?

2. Why only caring about $A_{t-1}^E$? I think $A_{t-2}^E$ can also cause leakage. Or can you prove that the former independence indicates the latter one?

**Limitations:**

As mentioned in Weaknesses, the current theories are not rigorous. They can be revised to make the whole logic consistent.

---

> ### Author Rebuttal · Authors · 2023-08-10
>
> We appreciate your constructive and insightful feedback.
>
> **1. Assumption on ideal imitator**
>
> Thank you for bringing up this important point. To clarify, in our problem setting of imitation learning from observation histories, we make the assumption that the control-relevant (state) information can be fully recovered from observation histories as we described in Abstract and Section 3.2. This assumption means that we consider only those POMDPs where the imitator policy can perform equally well as the expert policy, even when the imitator's action is determined solely by the observation history. Thus, under this assumption, the (marginal) action of an ideal imitator, denoted as $A^I_t$, would be exactly equal to $A^E_t$, satisfying Eq. (2). In such an ideal situation, the true value of CMI or HSCIC of our interest would indeed be 0.
> We acknowledge the importance of clarifying this assumption in detail to prevent any misunderstanding regarding our problem setting. In our next revision, we will provide a more explicit explanation of the assumption.
>
> **2. Relationship between loss and conditional dependence**
>
> While Theorem 1 serves the purpose of justifying the regularization of representations rather than the direct action output within our proposed PALR method, we understand the need for a more direct link between loss reduction and dependence minimization.
> Regarding the proposal for a new theorem, we recognize the complexity of establishing a strict relationship between lower loss and reduced dependence. The challenge arises from the fact that methods like FCA, MINE, and ours rely on individual estimations of dependence measures, and ensuring that lower loss strictly implies lower dependence necessitates accurate estimations across the board. However, existing literature demonstrates that minimizing estimated dependence measures, as observed in [A,B,C,D], has proven effective in practical scenarios.
> Consequently, while we acknowledge the theoretical difficulty in universally proving the connection, our approach aligns with established practices that demonstrate the practical utility of minimizing estimated measures of dependence. We have taken this approach to design our method and validated its effectiveness through empirical evaluations. We hope this clarification addresses the concern and illustrates the rationale behind our choice.
>
> **3. Performance degradation in `halfcheetah`**
>
> In our observation configuration, we’ve discovered that BC in `halfcheetah` task demonstrates competitive performance when provided with only a single observation. This suggests that the single observation contains sufficient information for effective control, potentially rendering additional historical context redundant in the scenario. In this situation, the remaining part of historical data may not significantly contribute to information for decision making.
>
> **4. Reason for caring only $A^E_{t-1}$**
>
> To maintain simplicity and unify the existing work into our framework, we focus on the one-step past action in this work. While we recognize the potential influence of $A_{t-2}^E$ or action histories, our primary objective is to show the efficacy of one-step past action regularization within IL-OH problems. The exploration of multi-step past action leakage naturally extends from our current work and we consider it a promising avenue for future research.
>
> We really appreciate that let us inform typos and important references and we update them in the revised version of our manuscript. We acknowledge the confusion caused by our use of the notation $\varphi_t$ for both random variables and values of representation. To enhance clarity in our presentation, we will introduce a distinct notation.
>
> **References**
>
> [A] Belghazi et al., “Mutual information neural estimation.”, ICML 2018.
>
> [B] Pogodin et al., “Efficient conditionally invariant representation learning.”, ICLR 2023.
>
> [C] Poole et al., “On variational bounds of mutual information.”, ICML 2019.
>
> [D] Quinzan et al., “Learning counterfactually invariant predictors.”, arxiv 2022.

---

> > ### Comment · Reviewer_4vtt · 2023-08-15
> > **Thanks for your rebuttal!**
> >
> > I appreciate the response from the authors and the additional experimental results. I think the authors' clarification is convincing and makes the paper sounder. I recommend accepting this paper.
> >
> > Here are some further suggestions:
> >
> > 1. Please describe the detailed experiment setup, environment, and model architecture of CARLA (if you want to put it into your final paper). Given that you have adopted a specific version of the CARLA environment in alignment with D4RL, distinct from the more intricate settings outlined in references [1, 2, 3],
> >
> > 2. I highly recommend adopting a consistent nomenclature for the problem under investigation. The existing literature contains varied terminologies denoting the same phenomenon, such as the inertia problem, copycat problem, latching problem, and leakage of past action, etc. To facilitate the advancement of this subject, it is advisable to standardize the terminology. Since you follow [4] closely, utilizing ''copycat problem" consistently throughout your paper could contribute to clarity and coherence.
> >
> > [1] Codevilla et al. ''Exploring the limitations of behavior cloning for autonomous driving."
> >
> > [2] Wen et al., ''Keyframe-Focused Visual Imitation Learning” , ICML 2021.
> >
> > [3] Wen et al., ''Fighting Fire with Fire: Avoiding DNN Shortcuts through Priming”, ICML 2022.
> >
> > [4] Wen et al., ''Fighting Copycat Agents in Behavioral Cloning from Observation Histories.", NeurIPS 2020.

---

> > > ### Author Response · Authors · 2023-08-15
> > > **Thank you for your follow-up feedback**
> > >
> > > We greatly appreciate your valuable comments.
> > > Following your suggestions, we will provide figures and tables that detail the overall network architecture, environment setup, and hyperparameters we used for CARLA tasks in the appendix of our next revision.
> > > In addition, we will ensure that we unify the terminology in our final draft.

---

### Author Rebuttal · Authors · 2023-08-10

# **General Response**

We sincerely thank the reviewers for their insightful and detailed comments. Below, we address key questions and feedback that have been consistently raised by the reviewers. If there are any aspects that still need clarification or elaboration, we are more than happy to address them during the author-reviewer discussion period.

## **1. Experiment in CARLA Environment (Table A)**

To demonstrate the effectiveness of our method on high-dimensional observation scenarios, we extended our evaluation to the CARLA environment, an image-based autonomous driving task. We only consider pixel images as observations, excluding other information like velocity or sensor data. Leveraging `carla-lane-v0` dataset provided in D4RL dataset [A], we implement the imitator policy built upon ResNet34 architecture, and the parameters of ResNet34 are fixed and only used for feature extraction from images. Detailed results are presented in Table A of the provided PDF file. Notably, PALR outperforms the baseline methods by a significant margin. This observation supports our conclusion that PALR can successfully enhance imitation performance in complex offline IL-OH problems.

## **2. Effectiveness on Decision Transformer Policy (Table B)**

To assess the efficacy of our method in regularizing policies with complex network architectures, we conducted additional experiments employing the Decision Transformer (DT) [B], one of the prominent offline RL methods. Given that reward information is absent in the offline IL dataset, we inserted a reward input of 0 into DT's structure to retain its original configuration. Our regularization approach, termed DT-PALR, was applied to the last hidden state of DT. We evaluated both the standard DT and DT-PALR across three POMDP versions of Mujoco tasks, consistent with the scenarios outlined in Table 1 of the paper.

The results are presented in Table B of the PDF file. With the exception of the `halfcheetah` task, it becomes evident that DT's performance lags behind that of BC, as demonstrated in Table 1 of the manuscript. This divergence might be attributed to DT's utilization of a larger network size, rendering it more susceptible to capturing spurious causal relationships in environments where access to complete states and rewards is restricted. Encouragingly, our results indicate a substantial enhancement in the performance of DT for the `hopper` and `walker2d` tasks when subjected to the DT-PALR method. This observation strongly suggests the adaptability of our approach to intricate models.

## **3. Comprehensive Quantitative Analysis (Table C, D)**

To enhance the reliability of the quantitative analysis provided in our manuscript, we comprehensively evaluate HSCIC scores ($\widehat{\mathrm{HSCIC}}^2(a^I_t, a^E_{t-1} | a^E_{t-1})$) across all 8 problem settings (see Table C in PDF file). The results demonstrate that our method consistently reduces the conditional dependence between $a^I_t$ and $a^E_{t-1}$ given $a^I_{t}$. Notably, PALR achieves lowest HSCIC scores in entire problem settings.

Moreover, to show a more robust assessment of conditional independence, we also estimate conditional mutual information (CMI) $\hat{I}(a^I_t; a^E_{t-1} | a^E_{t-1})$ that also measures conditional dependence of our interest. (see Table D in PDF file) As we described in Section 4.2.1, CMI can be decompose into two MI terms, $I(a^E_{t-1}; \varphi_t, a^E_t)$ and $I(a^E_{t-1}; a^E_t)$, we estimate them using MINE [C] respectively. The table shows that PALR consistently presents lower CMI estimates compared to BC across 6 out of 8 problem settings.

We will include these responses into the appendix of our revised paper. Once again, we extend our gratitude for the valuable feedback from the reviewers.

## **References**

[A] Fu et al., “D4rl: Datasets for deep data-driven reinforcement learning.”, arxiv 2020.

[B] Chen et al., “Decision Transformer: Reinforcement learning via Sequence Modeling.”, NeurIPS 2021.

[C] Belghazi et al., “Mutual Information Neural Estimation.”, ICML 2018.

---

> ### Author Response · Authors · 2023-08-14
> **Update in Experiments with Additional Baseline Methods**
>
> We have taken some of the reviewer's concerns into consideration and conducted additional experiments to provide a more comprehensive comparison with non-regularization-based methods. As suggested, we have included two new baselines [D,E] and extended the evaluation to cover all the tasks included in our manuscript and rebuttal (4 POMDP versions of Mujoco tasks and the CARLA environment).
> In addition, to make comprehensive evaluations in CARLA, we also conducted an experiment on `carla-town` whose dataset is provided by D4RL as in `carla-lane`. Here are the summarized results (all scores are normalized within a range of 0 to 100, covering from the random agent score to the expert score):
>
> | Task | W | BC | RAP | FCA | MINE | PALR | Keyframe [D] | PrimeNet [E] |
> | --- | --- | --- | --- | --- | --- | --- | --- | --- |
> | hopper | 2 | 32.47 ± 2.85 | 20.19 ± 1.38 | 31.89 ± 2.54 | 24.98 ± 1.89 | **42.01 ± 2.44** | 32.01 ± 1.86 | 29.98 ± 1.57 |
> |  | 4 | 47.65 ± 3.43 | 32.61 ± 2.62 | 36.90 ± 2.35 | 37.60 ± 3.14 | **58.39 ± 2.76** | 45.74 ± 0.95 | 45.31 ± 2.77 |
> | walker | 2 | 53.04 ± 2.69 | 15.82 ± 2.03 | 63.11 ± 2.69 | 58.62 ± 5.52 | **79.83 ± 2.29** | 49.97 ± 2.32 | 48.50 ± 3.25 |
> |  | 4 | 63.15 ± 6.28 | 25.39 ± 2.14 | **81.88 ± 3.26** | 68.71 ± 6.66 | **83.42 ± 5.43** | 77.37 ± 1.97 | **79.17 ± 3.30** |
> | halfcheetah | 2 | 74.08 ± 2.33 | 63.90 ± 2.14 | 78.24 ± 2.80 | 76.29 ± 1.87 | **86.44 ± 1.09** | 64.26 ± 1.41 | 61.47 ± 1.90 |
> |  | 4 | 68.35 ± 2.60 | 58.97 ± 2.66 | 69.89 ± 2.64 | 73.4 ± 2.35 | **79.05 ± 4.28** | 55.71 ± 4.14 | 45.51 ± 1.66 |
> | ant | 2 | 56.25 ± 3.45 | 44.05 ± 1.19 | 51.08 ± 2.19 | 53.88 ± 1.87 | **59.57 ± 3.03** | 54.94 ± 1.68 | 51.72 ± 2.38 |
> |  | 4 | **64.39 ± 1.77** | 48.63 ± 2.63 | 57.73 ± 1.25 | 56.56 ± 1.76 | **64.64 ± 2.53** | 48.59 ± 3.75 | 58.18 ± 1.92 |
> | carla-lane | 3 | 53.82 ± 7.66 | 20.15 ± 7.91 | 51.83 ± 7.91 | 60.62 ± 6.40 | **72.27 ± 2.62** | 66.36 ± 3.44 | 61.72 ± 1.53 |
> | carla-town | 3 | -3.94 ± 1.68 | -7.64 ± 0.97 | -9.42 ± 0.47 | **0.13 ± 0.93** | -1.15 ± 1.01 | -6.68 ± 0.07 | -9.50 ± 0.33 |
>
> For [D] (denoted as ‘Keyframe’ in the table), we used a softmax function as a weighting function of the weighted BC in Mujoco tasks and a step function in CARLA tasks, following the experiment of the original paper. Similar to regularization-based methods, we selected optimal hyperparameters $\tau \in [0.01, 0.1, 1, 10, 100]$, representing the softmax temperature for Mujoco tasks. For CARLA tasks, we used the same hyperparameters (threshold=0.1, weight=5) as in the CARLA experiment of the original paper.
>
> Regarding the `carla-town` task, none of the algorithms surpassed the random policy's performance (which scores 0), indicating a lack of meaningful imitation results. This outcome is likely due to the dataset containing numerous instances of the same observations paired with different actions (59,984 observation-action pairs out of 100K pairs in total).
>
> We plan to update these additional results into the revised version of our manuscript. We hope that these additional results can alleviate the concerns of reviewers.
>
> **References**
>
> [D] Wen et al., “Keyframe-Focused Visual Imitation Learning” , ICML 2021.
>
> [E] Wen et al., “Fighting Fire with Fire: Avoiding DNN Shortcuts through Priming”, ICML 2022.

---

### Decision · Program_Chairs · 2023-09-21

**Decision:**

Accept (spotlight)

**Comment:**

This is high-quality work that provides a novel method designed to address information leakage in imitation learning, which a problem of interest to the NeurIPS community. The authors should carefully revise the submission to include the additional results that they provided during the discussion period and also address several important points raised by the reviewers, including making more clear the assumptions made, revising notation for clarity, and explicitly acknowledging/explaining the limited experimental setting.